# Analytical Hysteretic Behavior of Square Concrete-Filled Steel Tube Pier Columns under Alternate Sulfate Corrosion and Freeze-Thaw Cycles

**DOI:** 10.3390/ma15093099

**Published:** 2022-04-25

**Authors:** Tong Zhang, Qianxin Wen, Lei Gao, Qian Xu, Jupeng Tang

**Affiliations:** 1School of Civil Engineering, Liaoning Technical University, Fuxin 123000, China; zt_1987_zt@163.com (T.Z.); 18242477225@163.com (Q.W.); 2School of Mechanics and Engineering, Liaoning Technical University, Fuxin 123000, China; jupengt@126.com; 3Liaoning Hanshi Technology Group Co., Ltd., Fuxin 123099, China; gao33lei33@163.com

**Keywords:** sulfate corrosion, freeze-thaw cycle, alternation effect, CFST pier column, hysteretic behavior

## Abstract

The hysteretic behavior of square concrete-filled steel tube (CFST) stub columns subjected to sulfate corrosion and freeze-thaw cycle is examined by numerical investigation. The constitutive model of steel considered the Bauschinger effect, and compression (tension) damage coefficient was also adopted for the constitutive model of core concrete. The experimental results are used to verify the finite element (FE) model, which could accurately predict the hysteretic behaviors of the CFST piers. Then, the effects of the yield strength of steel, compressive strength of concrete, steel ratio, axial compression ratio, and alternation time on ultimate horizontal load are evaluated by a parametric study. The results showed that the yield strength of steel and the steel ratio have a positive effect of hysteretic behavior. The compressive strength of concrete and alternation time significantly decreased the unloading stiffness which causes the pinching phenomenon. The yield strength of steel, compressive strength of concrete, and alternation time of environmental factors (corrosion-freeze-thaw cycles) has no obvious effect on the initial stiffness, while the steel ratio has a remarkable effect. The ultimate horizontal load increases with the increasing steel ratio, yield strength of steel and compressive strength of concrete. Meanwhile, the decrement of alternation time led to the increase of ultimate horizontal load. This suggests that the confinement coefficient and alternation time are the two main factors that impact the ultimate horizontal load. A formula which considers the reduction coefficient for the ultimate horizontal load of the CFST columns subjected to sulfate corrosion and freeze-thaw cycles is proposed. The formulae can accurately predict the ultimate horizontal load with mean value of 1.022 and standard deviation of 0.003.

## 1. Introduction

Concrete-filled steel tubes (CFST) are composite structural members with an outer steel tube and an inner core of concrete [1,2]. The utilization of CFST columns has been commonplace in recent decades due to the infilling concrete supporting the steel tube, which prevents its inward buckling and postpones its outward buckling [3]. Moreover, the steel tube restricts the core concrete in transverse, and thus, due to the confinement effect, the concrete is under three-dimensional compression, which leads to the improvement of load-bearing capacity and ductility [4,5]. Considerable research of CFST columns has been conducted on buildings and bridges’ rib arches [6,7,8,9]. In addition, the feasibility of using CFST columns as bridge piers has been reported [10,11,12,13].

Tomii et al. [14] firstly studied the seismic behavior of square CFST through a quasi-static method and illustrated that CFSTs were excellent in energy dissipation. K. Nakanishi et al. [15] investigated the peak horizontal load and ductility of bridge pier columns with different composite forms experimentally. The study theoretically verified the analogous rule between full-scale models and scaled-down specimens and found that the CFST pier columns were better in resistant cyclic loading. Aval et al. [16] adopted an inelastic fiber element model considering the axial elongation and curvature of steel tubes and concrete to analyze CFST columns under cyclic load. The results showed that the CFST with studs or reinforcement ribs has a higher dissipation of energy. M. Bruneau et al. [17] evaluated the limitation of existing design codes for circular CFST bridge piers by establishing an experimental database. Then, based on the simple plastic model, a new equation was proposed and the specifications “Guide for Load and Resistance Factor Desing (LRFD) Criteria for Offshore Structures” were recommended for the seismic design of bridge piers. Li [18] carried out a dynamic time history analysis of CFST piers according to practical engineering at a 7-degree (*M* = 0.58*I* + 1.5, where *M* is Richter scale, *I* denotes seismic fortification intensity, *I* = 7 here) protected earthquake intensity. It illustrated that the vertex displacement of pier columns was below the ultimate displacement for all 18 earthquake excitations, which verified that the CFST pier column has good seismic resistant. Wang [19] studied the effect of concrete fill percentage on the seismic behavior of CFST pier columns with filling ratios of 0%, 23%, 27%, 30%, and 31%. The study also considered the slenderness ratio and width-to-thickness ratio. It showed that the supporting role of concrete has the advantage of seismic resistance but the optional filling rate requires further investigation.

The bridge pier is generally used as a pressure-bearing member and works under complex conditions including load action, vehicle impact, harsh environment, and seismic effect, which takes disadvantage of the lifelong service of the bridge [20]. Chen et al. [21] investigated the effect of corrosion ratio and recycled aggregate replacement ratio on the axial compressive behavior of circular CFST columns by experimental and numerical research. It demonstrated that the type of recycled aggregate has a slight effect on loadbearing capacity of specimens under sulfate corrosion. Chen et al. [22,23,24] consider the effect of corrosion ratio and recycled aggregate replacement ratio to investigate the axial compressive behavior of 16 square CFST columns under sulfate corrosion experimentally and numerically. They demonstrated that the recycled aggregate type has negligible effect on specimens under sulfate corrosion. However, the corrosion time resulted in the decrement of ultimate bearing capacity, ductility, and composite elastic modulus of the specimen. FE models considering the reduction of thickness of the steel tubes could effectively simulate the effect of sulfate corrosion on steel tubes. Zhang [25] carried out a durability test on 20 thin-walled steel tube columns under SO_4_^2−^ corrosion. The study found that ultimate bearing capacity, ductility, and stiffness decreased with the corrosion ratio, especially for the specimens with thinner walls. The finial failure mode of the CFST stub columns under sulfate corrosion was due to shear failure. After corrosion by SO_4_^2−^, local bucking of the slender steel tube would occur to a large extent.

Yang et al. [26] experimentally dealt with the effect of freeze-thaw cycles, section type, and compressive strength of axially loaded CFST columns under freeze-thaw conditions. The test results demonstrated that the freeze-thaw cycle significantly influenced the loadbearing capacity, composite elastic modulus, and ductility index but slightly impacted the failure mode. They mentioned that the circular specimens performed better with regards to ductility. Wang [27] carried out an experimental test of axially loaded square CFST columns with an end plate to enlarge the effect of freeze-thaw cycles on compressive behavior. Wang [27] found that the cyclic time has a negative linear relationship on loadbearing capacity and initial stiffness of the specimens. Yan [28,29,30] investigated the axially loaded CFST stub columns under the freezing temperatures of −30 °C, −60 °C, and −80 °C. The test results presented that square specimens were destroyed in crushing failure, and few specimens appeared tensile fracture. Freezing temperature improved the ultimate compressive load and initial stiffness of CFST columns but decreased the ductility. Gao et al. [31] focused on the mechanical behaviors of circular CFST stub columns under freeze-thaw cycles in experimental tests. The effect of cyclic time and compressive strength of concrete for the axially loaded specimens was analyzed. Gao et al. [31] found that axial compressive strength decreased linearly with cyclic time while it increased with a higher compressive strength of concrete.

The durability of CFST pier columns is significantly influenced by a harsh environment, such as corrosion [32] and freeze-thaw [33]. However, few studies have investigated the compound influence of multiple environmental factors, which is in accordance with practical structure. Studies on dynamic properties were concentrated on CFST columns in ambient conditions. The steel tube suffering alternating sulfate corrosion and freeze-thaw cycles will result in higher brittleness and the concrete will crack at the interfacial transition zone due to the mechanical deterioration of the specimen. In addition, the seismic effect coupling with the alternation of sulfate corrosion and freeze-thaw cycles will rapidly cause material deterioration, and thus will cause corrosion fatigue cracks to occur more easily. This study thus carries out a simulation study of hysteretic behavior on 126 square CFST pier columns. Based on the reasonable constitutive model, nonlinear finite element (FE) models are set to analyze the CFST pier columns with a failure mode of shear failure by ABAQUS software. Then, the effect of strength of materials, steel ratio, axial compression ratio, and alternate time on the horizontal load-deformation hysteretic curve and skeleton curve. Ultimately, the reduction coefficient corresponding to the ultimate horizontal load was analyzed and a design formula to predict the ultimate horizontal load of square CFST pier columns after alternate of sulfate corrosion and freeze-thaw cycles was proposed.

## 2. Empirical Experimental Research

The square CFST pier columns were corroded through electrochemical corrosion. The electrolyte was a mixed solution of Ca(NO_3_)_2_, Na_2_SO_4_, and NH_4_Cl and adopted HNO_3_ to adjust pH value, which was set as 4.5. The specimens were fully immersed into the electrolyte. The corrosion rate was defined as the loss of mass of the steel and corrosion rate *γ* = 10%, 20%, and 30% corresponded to the terms of service of 5, 10, and 15 years. Hemispherical pitting was formed on the surface of the steel tubes. The test results reported in previous studies [34,35,36,37,38,39,40,41,42,43] were selected to verify the CFST pier columns modeled in ABAQUS software [44]. Table 1 summarizes their experimental test results. The parameters of each specimen were listed in Table 1, and it can be found that the specimen labels dividing different groups were the same as those tested in [34,35,36,37,38,39,40,41,42,43]. As shown in Table 1, the length of the specimens ranged from 530 to 1250 mm, corresponding to a length-to-width (*L*/*B*) ratio of 2.12–5.00. Thus, the influence of the slenderness effect could be avoided [45]. The compressive strength of the concrete cores ranges from 33.0 to 75.1 MPa, which includes normal-strength concrete and high-strength concrete [45] (*f*_ck_ ≤ 50 MPa is defined as normal strength concrete). Specimens with reinforcement bars consist of 8Φ16 and 4Φ11, and the ties used are Φ8@100 mm and Φ8@200 mm, which meet the specification requirements [46]. Then, the yield strength of the steel tubes varies from 242.2 to 389.0 MPa, and the axial compression ratio ranges from 0.2 to 0.60. The material properties of steel and infilled concrete can be obtained from tests or specifications such as GB50010 [46], ACI-318-08 [47], EN-1992-1-1 [48], GB50017 [49], AS4100 [50], etc. Figure 1 presents details of the typical testing device with which the hysteretic behavior of the specimens could be tested.

The method of quasi-static tests was generally adopted to investigate the effect of reciprocating cyclic loading on structural members. This method took advantage of easy operation and observation, controlled loading process, and convenience of checking data, and thus the quasi-static test method was selected to investigate the hysteretic behaviors of specimens. According to JGJ/T101-2015 [51], the loading history of cyclic loading included a force control phase and a displacement control phase, as showed in Figure 2. In the force control phase, one cycle was imposed at the load levels of 0.5*N*_vy_, 0.5*N*_vy_, and *N*_vy_, respectively, where *N*_vy_ is the estimated value of specimen yield strength. Afterward, displacement-controlled loading was performed at the incremental levels of 1*δ*_y_, 2*δ*_y_, 3*δ*_y_, etc, where *δ*_y_ represents the estimated displacement corresponding to the yield strength, and the cyclic loading was reported twice for each displacement level. The cyclic loading was repeated twice for each displacement level. The yield strength *N*_vy_ and yield displacement *δ*_y_ were tested with the GYM (general yield moment) method. Horizontal deformation was measured by linear variable differential transformers (LVDTs) and strain was measured by strain gauges. Final failure was defined as horizontal load decreasing to 85% of its peak value [51]. The test results of the specimens in Refs. [34,35,36,37,38,39,40,41,42,43] were listed in Table 1.

The test results obtained for the square CFST columns under reciprocating cyclic loading in terms of different main parameters were sketched and compared with the FE results in Section 3.5 in the forms of failure mode, skeleton curve, and horizontal ultimate compressive load in detail.

## 3. Finite Element Modeling

The phenomenological modeling of square CFSTs is typically performed by using differential [52,53] or non-differential models [54,55] and is determined by three main parts, including core concrete, steel tube, and interaction between these two raw materials. In addition, in FE analysis of the specimens, selection of element type and mesh size should balance the accuracy of the simulation results and simulation time. Since square CFST pier columns under cyclic load are not symmetrical, the whole specimen is modelled to investigate the hysteretic behaviors.

### 3.1. Material Properties

#### 3.1.1. Steel Tube

A secondary plastic flow stress-strain curve [45] was modified according to the effect of corrosion-freeze-thaw cycles on the relationship of stress–strain and then adopted to define the material behavior under sulfate corrosion and freeze-thaw cycles of steel tubes, which include five stages: elastic, elastic-plastic stage, plastic, yield strength, and strengthening and secondary plastic flow stages. The main parameters for defining the stress–strain curve of a steel tube include the limit of proportionality (*f*_p_), yield strength of the steel tube (*f*_y_), ultimate compressive strength (*f*_u_), and the corresponding strain (*ε*_e_, *ε*_e1_, *ε*_e2_ and *ε*_e3_). The five-stage stress–strain model of a steel tube can be defined as Equation (1) shows. Young’s modulus and Poisson’s ratio in the model adopted were 206 GPa and 0.283 [47,49] respectively.
(1)$$\sigma_{s}=\left\{ \begin{array}{ll} E_{\mathrm{se}}\varepsilon_{s} & \varepsilon_{s}\leq \varepsilon_{e}\\ -A\varepsilon_{s}^{2}+B\varepsilon_{s}+C & \varepsilon_{e}<\varepsilon_{s}\leq \varepsilon_{e1}\\ f_{\mathrm{ye}} & \varepsilon_{e1}<\varepsilon_{s}\leq \varepsilon_{e2}\\ f_{\mathrm{ye}}\left[1+0.6\frac{\varepsilon_{s}-\varepsilon_{e2}}{\varepsilon_{s3}-\varepsilon_{e2}}\right] & \varepsilon_{e2}<\varepsilon_{s}\leq \varepsilon_{e3}\\ 1.6f_{\mathrm{ye}} & \varepsilon_{s}>\varepsilon_{e3} \end{array}\right.$$
where *ε*_e_ = 0.8*f*_ye_/*E*_se_, *ε*_e1_ = 1.5*ε*_e_, *ε*_e2_ = 10*ε*_e1_, *ε*_e3_ = 100*ε*_e1_, *A* = 0.2*f*_y_(*ε*_e1_ − *ε*_e_)^2^, *B* = 2*Aε*_e1_, *C* = 0.8*f*_ye_ + *Aε*_e2_ − *Bε*_e_, *E*_se_ = (1 − 0.525*γ*)*E*_s_, *f*_ye_ = (1 − 0.908*γ*)*f*_y_, and *β* = Δ*t*/*t*, Δ*t* = *t* − *t*_e_.

When the CFST columns were subjected to the alternation effect of sulfate corrosion and freeze-thaw cycles, not only the thickness, but also the mechanical properties of the steel tube changed. A series of typical stress–strain curves of steel tubes of 345 MPa with corrosion rates of 0%, 10%, 30%, and 50% was presented in Figure 3. The von Mises criterion is adopted to estimate steel yield stress.

The Bouc–Wen modified model [56] is not particularly sensitive to the selection of its parameters. Vaiana et al. [57] proposed a hysteretic mechanical system by combining a novel rate-independent model and an explicit time integration method which avoid the convergence problem for its small time step. Since the hysteretic behaviors of steel tubes under cyclic loading are complex and the effects of corrosion-freeze-thaw cycles have strong randomness, using a bilinear model is more easily describes the complex environment effects. The stress–strain skeleton curve of a steel tube corresponding to its hysteretic behavior is composed of two parts, which are the elastic stage and the strain hardening stage, as shown in Figure 4. It can be seen that unloading occurred in the strain hardening stage, which caused non-negligible residual strain. The residual stress would decrease when reverse loading, which caused dimensional stability issues. Due to the effect of residual strain on the loading and unloading process, the Bausinger effect should be taken into account. The stress–strain model of the steel tube was presented in Equation (2), where the elastic modulus in the strain hardening stage was defined as 0.01*E*_se_.
(2)$$\sigma =\left\{ \begin{array}{ll} E_{\mathrm{se}} & \varepsilon \left(0\leq \varepsilon \leq \varepsilon_{y}\right)\\ f_{\mathrm{ye}}+0.01E_{\mathrm{se}}\left(\varepsilon -\varepsilon_{y}\right) & \left(\varepsilon \geq \varepsilon_{y}\right) \end{array}\right.$$
where *σ* is the stress of the steel tube, *E*_se_ is the elastic modulus of steel, *ε* is the strain of the steel tube below yield strain, *ε*_y_ is the yield strain corresponding to the yield strength of the steel tube.

#### 3.1.2. Confined Core Concrete

The stress–strain model proposed by Han et al. [45] was adopted to simulate the properties of core concrete. The material law for the core concrete under the alternation effect of sulfate corrosion and freeze-thaw cycles was illustrated in Figure 5.

The core concrete is modelled through the concrete damage plasticity (CDP) model. A stress–strain model considering the confinement effect in improving the behavior of core concrete by Han et al. [45] was applied to define the concrete material in the FE model.
(3)$$y=\left\{ \begin{array}{ll} 2x-x^{2} & \left(x\leq 1\right)\\ \frac{x}{\beta {\left(x-1\right)}^{\eta }+x} & \left(x>1\right) \end{array}\right.$$
where *x* = *ε*/*ε*_0_, *ε* is the strain of the concrete and *ε*_0_ is the peak strain corresponding to peak load, *ε*_0_ = (1300 + 12.5$f_{c}^{\prime }$ + 800*ξ*_e_^2^) × 10^−6^, *y* = *σ*/*σ*_0_, *σ* is the stress of concrete and *σ*_0_ is the stress corresponding to the peak load, *σ*_0_ = *f*_c_^’^(1 − 0.035*N*_ft_/100), $f_{c}^{\prime }$ = [0.76 + 0.2log_10_(*f*_cu_/19.6)]*f*_cu_, $f_{c}^{\prime }$ is the cylinder strength of concrete, *f*_cu_ is the cube compressive strength of concrete, *η* = 1.6 + 1.5/*x*, *η* is defined as shape coefficient to reflect the properties of shape cross section, *β* = ($f_{c}^{\prime }$)0.1/[1.2(1 + *ξ*_e_)0.5], *ξ*_e_ =*α*_e_ × *f*_ye_/*f*_ck_, *ξ*_e_ is the effective confinement factor, *f*_ye_ and *f*_ck_ are the effective yield strength of steel tube and characteristic compressive strength of concrete, respectively; *f*_ck_ = 0.88 × 0.76 × *f*_cu_, *α*_e_ = *A*_se_/*A*_c_; *α*_e_ is the steel ratio, and *A*_se_ and *A*_c_ are the effective cross-sectional areas of the steel tube and core concrete, respectively.

The elastic modulus of core concrete was calculated by Equation (4), which was proposed by ACI 318 [47] as follows:(4)$$E_{c}=4700\sqrt{f_{c}^{\prime }}$$
where *E*_c_ is the elastic modulus of core concrete and $f_{c}^{\prime }$ denotes the cylinder strength of concrete.

The parameters of the CDP model were defined in the numerical model using the formula provided by Tao et al. [58]. The value of the dilation angle (*ψ*), *K*_c_, *f*_b0_/*f*_c0_ was determined by Equations (5)–(7). The eccentricity value was selected as 0.1 and the viscosity parameter was chosen as 0.0001 to improve the convergency of the model.
(5)$$\psi =\left\{ \begin{array}{ll} 56.3\left(1-\xi_{e}\right) & \xi_{e}\leq 0.5\;\\ 6.672e^{\frac{7.4}{4.64+\xi_{e}}} & \xi_{e}>0.5 \end{array}\right.$$
(6)$$K_{c}=\frac{5.5}{5+2{\left(f_{c}^{\prime }\right)}^{0.075}}$$
(7)$$f_{b0}/f_{c0}=1.5{\left(f_{c}^{\prime }\right)}^{0.075}$$
where ξ_e_ is the effective confinement factor, $f_{c}^{\prime }$ is the cylinder strength of concrete, *f*_b0_ is the initial biaxial strength of concrete, and *f*_c0_ is the initial axial strength of concrete.

Since the stress–strain relationship of concrete under hysteretic load could be replaced by that under uniaxial load [45], the stress–strain model of uniaxial load was adopted herein to be the skeleton curve of the hysteretic relationship. However, the concrete of the CFSTs under hysteretic load experienced the phenomenon of strain softening and stiffness degradation, which were caused by the development of cracks. Based on the characteristics of energy absorption and energy dissipation in the process of plastic deformation, the concrete damage model proposed by Britel and Mark [59] was introduced to the constitutive model of core concrete
(8)$$D=\frac{W_{0}-W_{\varepsilon }}{W_{0}}=\frac{W_{0}-\sum_{j=1}^{a}W_{\varepsilon j}}{W_{0}}$$
where *D* is the damage variable of core concrete, *W*_0_ presents the strain energy in nondestructive state, *W*_ε_ presents the strain energy under damage state, and *a* denotes the number of calculation steps.
(9)$$d_{c\left(t\right)}=1-\frac{\sigma_{c\left(t\right)}E_{c}^{-1}}{\varepsilon_{c\left(t\right)}^{\mathrm{pl}}\left(1/b_{c\left(t\right)}-1\right)+\sigma_{c\left(t\right)}E_{c}^{-1}}$$
where *σ*_c(t)_ is the compression (tensile) stress under a damaged state and *E*_c_ denotes the elastic modulus of concrete. In order to ensure the convergence and rationality, coefficient *b*_c(t)_ is selected as 0.5 and 0.25 corresponding to compression and tension condition, respectively.

The tension recovery of core concrete should be considered in the CDP model under low cyclic load. This characteristic was determined by compression stiffness recovery parameter *w*_c_ and tension stiffness recovery parameter *w*_t_. According to a former study, micro cracks in concrete could convey compression under compressive load, while concrete with micro cracks could not convey tensile strength [60]. In this model, *w*_c_ = 1.0 and *w*_t_ = 0 were employed.

### 3.2. Element Type and Mesh

According to a former study [61], core concrete was modeled using 3D solid element (C3D8R) with reduced integration. The steel tube was modelled using a 4-node 3D shell element (S4R), where 9 integrations were set along the thickness of shell elements using the Simpson integration rule [62]. The steel bar selected was a three-dimensional two-node element (T3D2). All of the element types are available in the ABAQUS program library. The whole structural meshing technic in ABAQUS was adopted to obtain an evenly distributing mesh style. A form of the model in simulation was presented in Figure 6. The element size in longitudinal and lateral directions was selected to be *B*/10 for both the core concrete and the steel tube. The 3D model consisted of a mesh which was deemed fine enough to accurately calculate the hysteretic behaviors of CFSTs with a reasonable computational time.

### 3.3. Boundary Condition and Load Application

The centers of the top and bottom surfaces were set as reference point 1 (RP1) and reference point 2 (RP2), respectively. The top and bottom surfaces of the specimen were constrained with RP1 and RP2 through a rigid body constraint. The top surface of the CFST column is free in all directions except in the loading direction. The axial load was exerted on the RP1 in the Z direction and the horizontal load was also exerted on RP1 in the X direction in the displacement control mode as shown in Figure 6 and Figure 7. The bottom surface of the specimen is restrained in all degrees of freedom. The boundary condition in the model can simulate the CFST column effectively, which corresponds to one end hinged and one end fixed in reality as shown in Figure 8, where *P* presents the lateral load, Δ_m_ denotes lateral displacement, and *L*_0_ defined as calculation length.

The model includes three steps: (1) an initial analysis step, where boundary conditions were set up; (2) step 2, where axial load is applied in the load-control mode; (3) step 3, where horizontal load exerted to RP1 with desired displacement in the displacement-control mode. The time of each analysis step was separated according to the displacement loading amplitude curve. The calculated result was obtained by Gauss integral iteration and output at the end of each analysis step.

### 3.4. Concrete-Steel Tube Interface Modeling

The concrete core and steel tube employed surface-to-surface contact to ensure the concrete core and the steel tube maintained an adequate bond, which created the CFST column as a whole. Since the stiffness of the concrete core was higher than that of the steel tube, the concrete core was selected as the master surface, while the steel tube was set as the slave surface. Moreover, the formula of small sliding was adopted in this section since the relative slip of the concrete core and the steel tube was small. Mechanical properties of tangential behavior were simulated by a penalty and friction coefficient, which is 0.6 [62]. The value of the shear stress limit was selected as 0.15 MPa [63]. Normal behavior adopted hard contact, which resulted in surface contact with no limitation.

### 3.5. Verification of Finite Element Model

#### 3.5.1. Failure Mode

The typical failure mode of specimens under cyclic loading at ambient temperature (20 ± 2 °C) was depicted in Figure 9. As shown in the figure, the FE model presents a shear failure mode with outward buckling at the end of the steel tube. The results of the FE models and test specimens were consistent, which suggests that the FE model was effective.

#### 3.5.2. Skeleton Curve

The experimental results were compared with those obtained via FE modelling in terms of skeleton curve, as depicted in Figure 10. It can be seen in the figure that in the elastic stage, the skeleton curves of specimens in accordance with the FE models.

However, in the elastic plastic stage, the slope of FE results was higher than test results. For the twelve verified specimens, the skeleton curves match the FE curves with a subtle difference in slope, which is likely to be associated with a geometric and loading defect generated through the experiment. These results were then employed in investigating hysteretic behavior.

#### 3.5.3. Horizontal Ultimate Compressive Strength

The horizontal ultimate compressive strength was obtained from the FE model and compared with the test specimens. Table 2 compares the peak load of the specimens with the capacity derived from the FE modelling. The mean and standard deviation values highlight a favorable efficiency of the FE model in predicting the desired measures.

#### 3.5.4. Hysteretic Behavior of CFSTs after Sulfate Corrosion

The experimental results were compared with the results of the FE model in terms of hysteretic behavior, as depicted in Figure 11. The rationality of the FE model on calculating ultimate horizontal load was verified by the existing test results [64,65,66], as shown in Figure 12.

## 4. Analysis and Discussion

In this paper, the typical specimen had a side length (*B*) of 400 mm, shear span ratio (λ) equal to 1.5, and other parameters as tabulated in Table 3. The number of cycles consisted of 100 of freeze-thaw cycles and 10% sulfate corrosion.

### 4.1. Yield Strength of Steel Tube

Figure 13 presents the effects of yield strength of a steel tube on hysteretic curves and skeleton curves of CFST stub columns under horizontal load after an alternation of sulfate corrosion and freeze-thaw cycles. It can be seen that the effect of yield strength of the steel tube was significant on the shape of the curves and cyclic displacement corresponding to ultimate bearing capacity. Regarding the hysteretic curves, with the increase of the yield strength of the steel tube, they became fuller, which resulted in a larger surrounded area. When the hysteretic curve enters the second half, the cyclic displacement declined faster with the larger yield strength of the steel tube due to the damage accumulation of the steel tube under cyclic reciprocating load after sulfate corrosion.

Skeleton curves presented in Figure 14 kept a similar trend in the elastic stage, which illustrated the effect of yield strength of the steel tube on initial stiffness. However, the effect of yield strength of the steel tube on horizontal load at peak point was significant, especially when the yield strength of the steel tube increased from 235 to 335 MPa. The ultimate horizontal load increased by 26.3%, 35.0%, and 39.2%, respectively as the yield strength of the steel tube increased from 235 to 335, 390, and 420 MPa. Meanwhile, the effect of yield strength of the steel tube on the slope of the curves in the descending stage and ductility was negligible.

### 4.2. Compressive Strength of Core Concrete

Figure 15 and Figure 16 depict the effects of the compressive strength of concrete on hysteretic curves and skeleton curves of CFST stub columns under horizontal load after an alternation of sulfate corrosion and freeze-thaw cycles. It can be seen that the effect of the compressive strength of concrete was significant on the shape of the curves and cyclic displacement corresponding to ultimate bearing capacity. For the hysteretic curves, when the compressive strength of the concrete increased, the stiffness of the specimen was weaker, which resulted in the “S” shape of the *P*-Δ_m_ curve. For skeleton curves, all four curves showed a similar trend in the elastic stage which illustrated that the effect of the compressive strength of the concrete on initial stiffness was slight. However, the effect of the compressive strength of the concrete on horizontal load at peak point was significant. The ultimate horizontal load increased by 16.0%, 27.7%, and 33.7%, respectively as the compressive strength of concrete increased from 30 to 40, 50, and 60 MPa. The compressive strength of concrete had some effect on the slope of the curves in the descending stage. With the increase of the compressive strength of the concrete, the descending stage of the skeleton curve decreased faster, which demonstrated that the ductility of the specimens was weaker due to the brittleness of the high strength of the concrete core. Although high compressive strength of concrete could improve the horizontal ultimate bearing capacity to some degree, the ductility would decrease, which should be focused on.

### 4.3. Steel Ratio

Figure 17 and Figure 18 depicted the variations of hysteretic curves and skeleton curves with different steel ratios under alternating sulfate corrosion and freeze-thaw cycles. The effects of steel ratio on hysteretic curve, skeleton curve, and bearing capacity corresponding to each cycle were significant. With the increase of steel ratio, the hysteretic curves changed from flat to full and the area within the curves became larger, which illustrated that the specimens were sufficient in the plastic deformation capacity.

It can be seen in Figure 18 that the slope of the four skeleton curves in the elastic stage was obviously different. The higher steel ratio resulted in higher elastic stiffness and ultimate horizontal load. The ultimate horizontal load increased by 21.4%, 61.0%, and 126.4% when the steel ratio increased from 0.06 to 0.10, 0.18, and 0.20, respectively. Moreover, the skeleton curves decreased slowly and even presented strain hardening phenomenon due to the confinement effect of the steel tube, which presented ductile behavior.

### 4.4. Axial Compression Ratio

Figure 19 and Figure 20 presented the effects of the axial compression ratio on the hysteretic curves and skeleton curves of CFST stub columns under horizontal load after an alternation of sulfate corrosion and freeze-thaw cycles. All the shapes of hysteretic curves demonstrated that the effect of the axial compression ratio was slight. The ultimate horizontal load slightly increased by 4.2%,6.6%, 8.1%, 9.0%, and 9.3%, respectively as the axial compression ratio increased from 0.05 to 0.1, 0.15, 0.20, 0.25, and 0.30 when the alternation time was three. The descending stage of the skeleton curves decreased more with the increase of the axial compression ratio, which demonstrated that the ductility of the specimens was weaker. This was for the reason that the material damage became more serious with the increasing axial compressive load due to the increasing axial compression ratio.

### 4.5. Sulfate Corrosion and Freeze-Thaw Cycles

Figure 21 depicted the effects of alternation time on hysteretic curves of CFST piers under horizontal load after an alternation of sulfate corrosion and freeze-thaw cycles. It can be seen that the effect of alternation time was significant on the shape of the curves and cyclic displacement corresponding to ultimate bearing capacity. For hysteretic curves, with the increase of alternation time, the phenomenon of pinch occurred, which resulted in the smaller area.

For skeleton curves as showed in Figure 22, five curves kept a similar trend in the elastic stage, which illustrated that the effect of alternate time on initial stiffness could be ignored. However, the effect of alternation time on the horizontal load at peak point was significant. The ultimate horizontal load decreased by 16.2%, 29.8%, 40.5% and 50.2%, respectively as the alternation time increased from 1 to 2, 3, 4, and 5. Meanwhile, alternation time has effects on the slope of the curves in the descending stage. The skeleton curve has a lower ductility, as the skeleton curves decreased faster in the descending stage due to the serious material damage caused by the alternate time on load resistance.

## 5. Simplified Formula for Horizontal Ultimate Compressive Strength

According to the parameter analysis in Section 4, the reduction factor of the horizontal ultimate compressive load of CFST under alternation between sulfate corrosion and freeze-thaw cycles is mainly determined by confinement coefficient (*ξ*) and alternation time (*i*). In this study, according to practical engineering, the range of parameters was selected as *ξ* = 0.6–2.1 and *i* = 0–5.

### 5.1. Design Method

According to Figure 7, the bending moment of CFST stub columns under low cycle reciprocating load was expressed in Equation (10) as follows:(10)$$M_{u}=P_{u0}L+N\Delta$$
where *M*_u_ represents the flexure capacity; *P*_u0_ denotes the horizontal ultimate axial load; *L* is the length of the CFST column; *N* stands for the axial compressive load; and *Δ* corresponds to lateral deformation.

Based on CJJ166-2011 [67], the axial compression ratio of CFST pier columns with regular section in earthquake fortification zone should be less than 0.3, i.e., *n* ≤ 0.3. Meanwhile, lateral deformation was far less than the calculated length of CFST pier column, so the bending moment caused by the axial compressive load could be ignored. Thus, the ultimate horizontal load was expressed in Equation (11) as follows:(11)$$P_{u0}=M_{u}/L_{0}$$

The design formula to calculate the flexure capacity in GB50936-2014 was described in Equations (12)–(14) as follows:(12)$$M_{u}=\gamma_{m}W_{\mathrm{sc}}f_{\mathrm{sc}}$$
(13)$$W_{\mathrm{sc}}=r_{0}^{3}/4$$
(14)$$\gamma_{m}=-0.483\xi +1.962\sqrt{\xi }$$
where *γ*_m_ represents the plastic adaption coefficient of cross section; *W*_sc_ denotes the modulus of section of CFST pier columns (mm^3^); *r*_0_ corresponding to the equivalent circle radius (mm); and *M*_u_ is the flexure capacity (*N*·mm).

### 5.2. Modified Reduction Coefficient

To evaluate reduction of loading bearing capacity of the square CFST pier columns after alternate sulfate corrosion and freeze-thaw cycles, the reduction coefficient *k*_sq_ was defined in Equation (15) as follows:(15)$$k_{\mathrm{sq}}=\frac{P_{\mathrm{ue}}}{P_{u0}}$$
where, *P*_ue_ is the ultimate horizontal load under sulfate corrosion and freeze-thaw cycles; *P*_u0_ is the ultimate horizontal load under ambient condition. The major effect was presented in Figure 23.

The reduction coefficient varied with alternation time (*i*) linearly. With the increase of alternation time, the reduction coefficient decreased rapidly and decreased up to 75.0% compared to the specimens under ambient conditions, which illustrated that *i* was the major factor that affects the ultimate horizontal load. In Figure 23a, the reduction coefficient decreased with the steel ratio significantly. The reduction coefficient decreased up to 64.0%, 61.0%, 59.1%, 59.0%, 53.1%, and 50.7%, respectively as the steel ratio increased from 0.06 to 0.08, 0.10, 0.15, 0.18, and 0.20, with the alternation time of 3. Thus, the steel ratio was selected to be a major factor of the reduction coefficient. 

It can be seen in Figure 23b,c the reduction coefficient decreased greatly with different compressive strengths of core concrete and yield strengths of steel tube with increasing alternation time. The specimens with lower compressive strength of core concrete or higher yield strength of steel tube corresponded to smaller reduction coefficients. The reduction coefficient decreased up to 61.6%, 59.1%, 59.0% and 56.4%, respectively corresponding to compressive strengths of core concrete of 50, 40, and 30 MPa, compared to the specimens with a compressive strength of core concrete of 60 MPa. Meanwhile, *k*_sq_ decreased up to 59.1%, 56.0%, and 52.3%, respectively as the yield strength of steel tube increased from 235 to 335, 390, and 420 MPa. 

Then, the reduction coefficient decreased obviously with the increasing confinement coefficient, as shown in Figure 23d. Since the confinement coefficient (*ξ* = *αf*_y_/*f*_c_) was determined by the steel ratio and the relative ratio of yield strength of the steel tube to compressive strength of the core concrete, the reduction coefficient considering *ξ* could reflect three above parameters. As shown in Figure 23e, the curves almost overlapped with the axial compression ratio (*n*) when it ranged from 0.05 to 0.30. This illustrated that the effect of axial compression ratio was negligible. Therefore, alternation time and confinement coefficient were adopted to be the major factors for evaluating the reduction coefficient of square CFST pier columns after alternate sulfate corrosion and freeze-thaw cycles. The reduction coefficient was described in Equation (16) as follows:(16)$$k_{\mathrm{sq}}=1-\left(0.008\xi^{2}+0.004\xi +0.119\right)i$$
where *ξ* is the confinement coefficient; *i* presents the alternate time.

### 5.3. Verified Design Model

The reduction coefficient of the horizontal ultimate compressive load of square CFST pier columns after sulfate corrosion and freeze-thaw cycles was calculated based on Equations (15) and (16). All the reduction coefficients corresponding to FE models were compared with the calculated results, as shown in Figure 24. It can be seen that the formula for predicting reduction coefficient has high accuracy with a mean value of 1.028 and a standard deviation of 0.001.

For the square CFST pier column after alternate of sulfate corrosion and freeze-thaw cycles, ultimate horizontal load can be calculated by modifying the formula under ambient condition, which is expressed as follows: (17)$$P_{\mathrm{be}}=k_{\mathrm{sq}}P_{u0}=\left[1-\left(0.008\xi^{2}+0.004\xi +0.119\right)\left(0.00375N_{ft}+6.25\gamma \right)\right]P_{u0}$$
where *k*_sq_ is the reduction coefficient; *P*_u0_ presents the ultimate horizontal load under ambient conditions; the relationship between *N*_ft_ and *γ* was defined as *i* in Ref. [68].

The comparison between the calculated values and numberical values of the ultimate horizontal load of square CFST pier columns after alternate sulfate corrosion and freeze-thaw cycles was presented in Figure 25. The error of the modified simplified formula in GB50936-2014 [62] was within 15%, with a mean value of 1.022 and standard deviation of 0.003, which illustrated that the proposed model was accurate in predicting the ultimate horizontal load. Therefore, the modified formula for GB50936-2014 [69] could be adopted to evaluate the ultimate horizontal load of square CFST pier columns after alternate sulfate corrosion and freeze-thaw cycles.

It should be mentioned that the modified simplified formula could be adopted to calculate the ultimate horizontal load of square CFST pier columns after alternate sulfate corrosion and freeze-thaw cycles with the limited parameters: (1) the section side length ranges from 200–1200 mm; (2) compressive strength ranges from 30–60 MPa; (3) yield strength of steel tubes ranges from 235–420 MPa, (4) alternation time ranges from 0–500; and (5) corrosion ratio ranges from 0%–50%.

## 6. Conclusions

The study presented a nonlinear finite element (FE) model to analyze the hysteretic behavior of concrete-filled steel tube (CFST) pier columns subjected to alternate of sulfate corrosion and freeze-thaw cyclic. Quantitative and qualitative verifications performed in this paper indicate that the FE model can accurately predict the hysteretic behavior of CFST pier columns. According to the numerical analysis, the following conclusions can be drawn:(1)The nonlinear finite element (FE) model considering the dual environmental impact of sulfate corrosion and freeze-thaw cycles was accurate in simulating the degradation of loading stiffness and unloading stiffness, bearing load, and ductility on the basis of reasonable damage factors.(2)The steel ratio, yield strength of steel tube, compressive strength of core concrete and alternation time have a significant effect on hysteretic behaviors of square concrete-filled steel tube (CFST) pier columns after alternate sulfate corrosion and freeze-thaw cycles. The cyclic curves of hysteresis loops increased with increasing steel ratio and yield strength of steel tube caused the increasing initial stiffness of the specimens. The skeleton curves corresponding to the hysteresis loops presented ductile behavior with increasing steel ratio and yield strength of steel tube. The effect of yield strength of the steel tube, compressive strength of core concrete, and alternation time on initial stiffness was negligible. The ultimate horizontal load increased by 21.4–126.4%, 26.3–39.2%, and 16.0–33.7% as the steel ratio increased from 0.06 to 0.20, the yield strength of steel tube increased from 235 MPa to 420 MPa, and compressive strength of core concrete increased from 30 MPa to 60 MPa, respectively while the ultimate horizontal load decreased by 16.2–50.2% as alternation time increased from 1 time to 5 times.(3)The steel ratio, compressive strength of core concrete, axial compression ratio, and alternation time have a significant effect on ductility behavior. The ductility of the specimens increased with the increasing steel ratio since the post stage of the skeleton curve varied from the descending stage to the ascending stage, while the ductility of the specimens decreased with the increasing compressive strength, axial compression ratio, and alternation time for the increasing brittleness caused by the damage of raw materials.(4)The confinement coefficient and alternation time were the two major factors that affect the reduction coefficient of the ultimate horizontal load of square CFST pier columns after alternate sulfate corrosion and freeze-thaw cycles. The horizontal bearing capacity decreased by 16.2–50.2% with the alternation time increasing from 1 time to 5 times. The horizontal bearing capacity decreased up to 17.7% as the confinement coefficient increased from 0.63 to 2.06.(5)A simplified formula for predicting the reduction coefficient of the horizontal bearing load with high accuracy with a mean value of 1.028 and standard deviation of 0.001 was developed.(6)A modified equation for predicting the ultimate horizontal load of square CFST pier columns after alternate sulfate corrosion and freeze-thaw cycles was proposed within the error of 15%.

## Figures and Tables

**Figure 1 materials-15-03099-f001:**
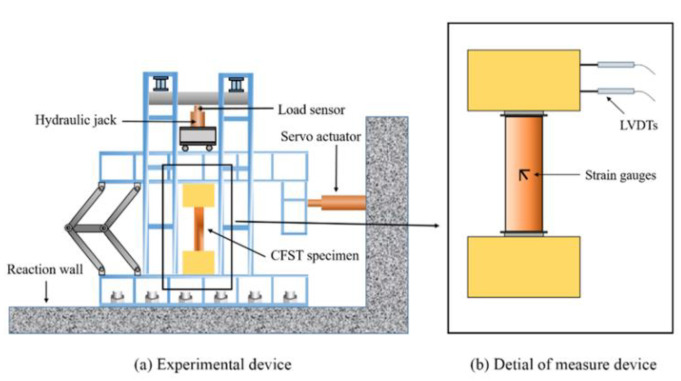
Experimental device and layout of the specimen.

**Figure 2 materials-15-03099-f002:**
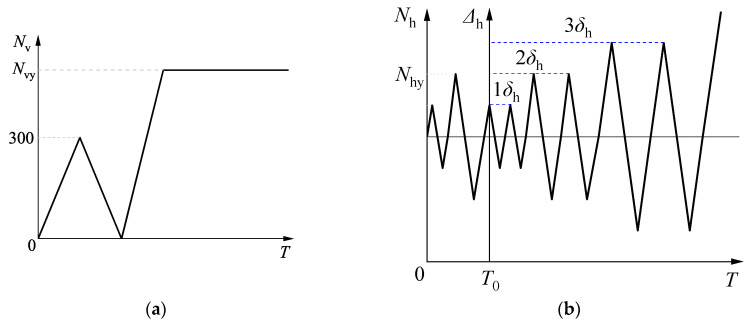
Loading system of specimens. (**a**) Vertical loading system; (**b**) Horizontal loading system.

**Figure 3 materials-15-03099-f003:**
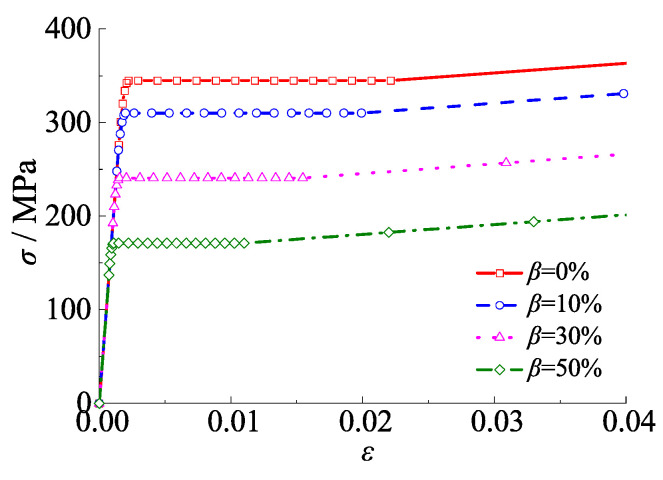
Stress–strain curve of steel tube under alternation of sulfate corrosion and freeze-thaw cycles.

**Figure 4 materials-15-03099-f004:**
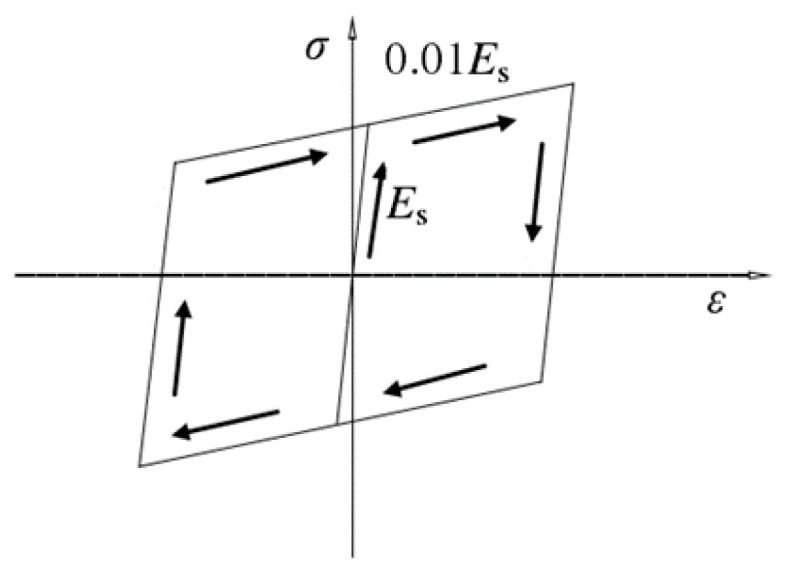
Bilinear kinematic hardening model of steel tube.

**Figure 5 materials-15-03099-f005:**
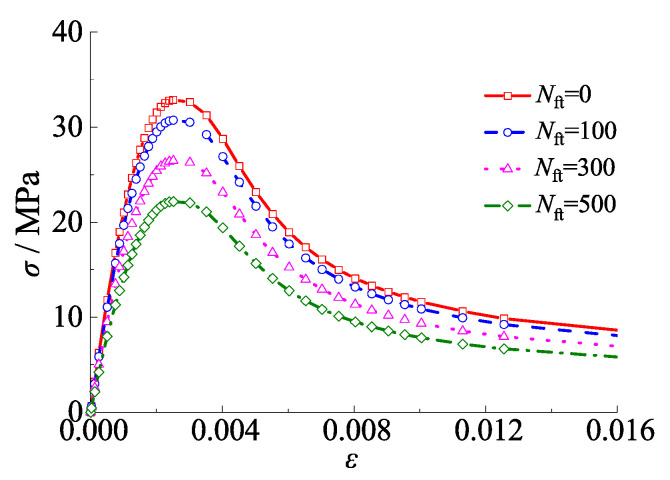
Stress–strain model for core concrete in square CFST stub columns subjected to alternating sulfate corrosion and freeze-thaw cycles.

**Figure 6 materials-15-03099-f006:**
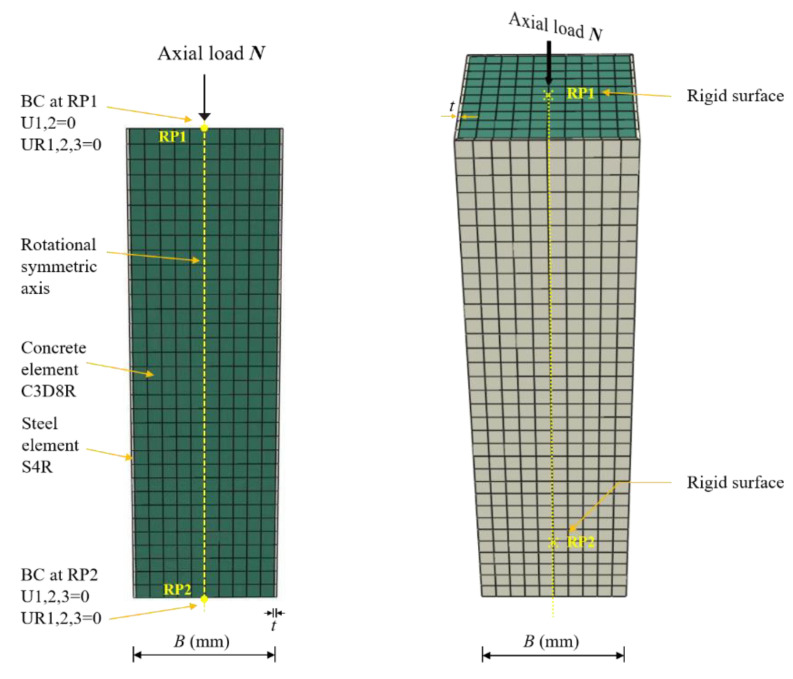
Schematic views of FE meshing and boundary conditions.

**Figure 7 materials-15-03099-f007:**
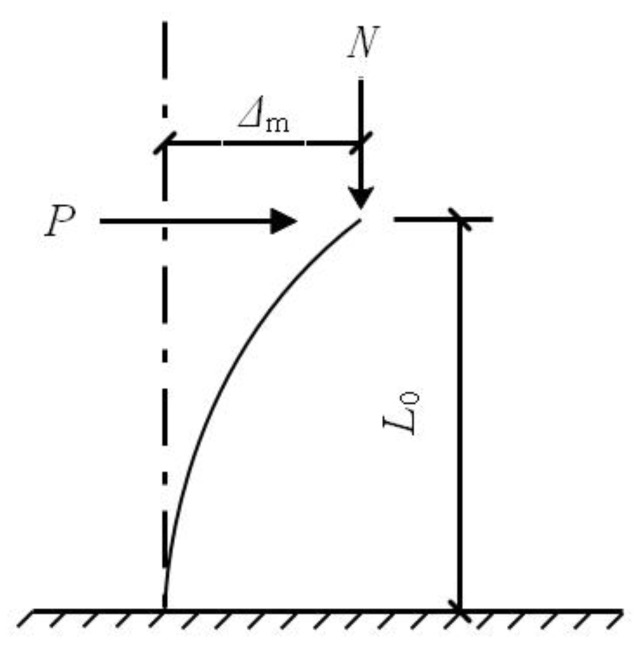
Schematic views of CFST stub columns under cyclic load.

**Figure 8 materials-15-03099-f008:**
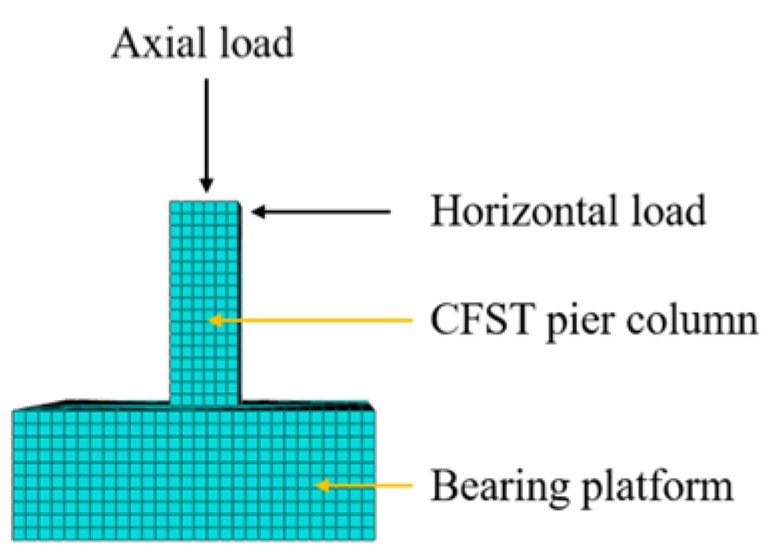
Schematic views of FE model under cyclic load.

**Figure 9 materials-15-03099-f009:**
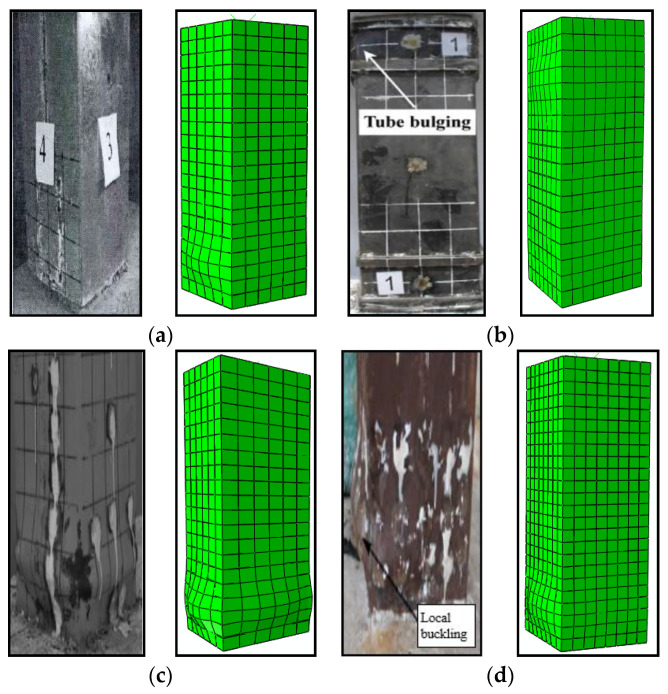
Comparison of failure modes between test specimens and FE models under cyclic loading under ambient temperature. (**a**) STRC-60-1.5-4; (**b**) STRC-70-1.5-4; (**c**) FGZ4; (**d**) A1.

**Figure 10 materials-15-03099-f010:**
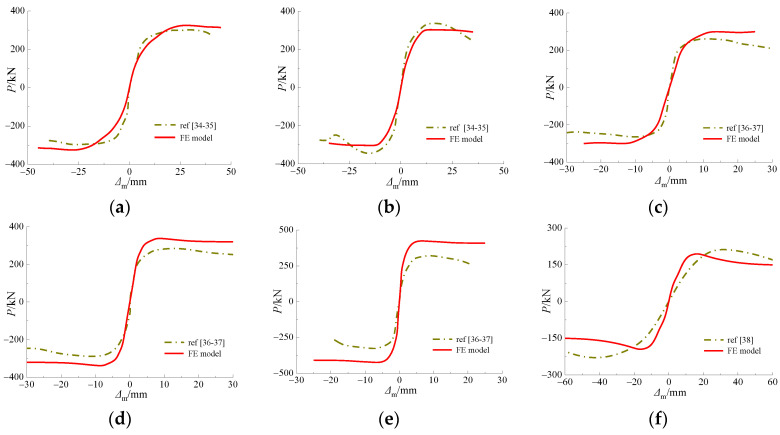
Comparison of skeleton curve between test specimen and FE model under cyclic loading at ambient temperature. (**a**) STRC-60-1.5-4; (**b**) STRC-60-1.5-6; (**c**) STRC-70-1.5-4; (**d**) STRC-70-1.5-5; (**e**) STRC-70-1.5-6; (**f**) FGZ4; (**g**) A1; (**h**) A2; (**i**) A3; (**j**) RS-4L; (**k**) RS-6M; (**l**) RS-10H.

**Figure 11 materials-15-03099-f011:**
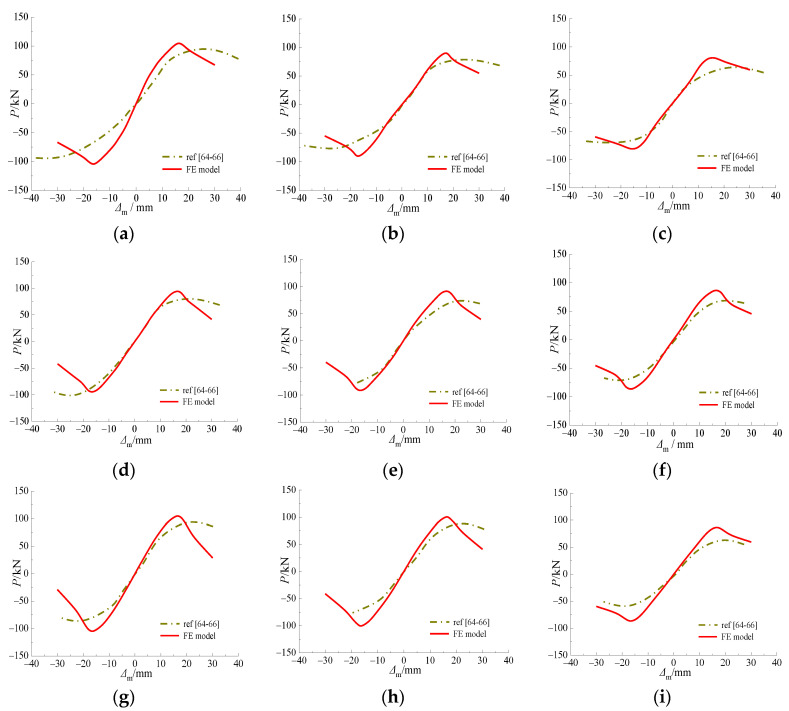
Comparison between FE model and test results of specimens after sulfate corrosion. (**a**) RC-0A; (**b**) RC-10A; (**c**) RC-20A; (**d**) RC-0B; (**e**) RC-10B; (**f**) RC-20B; (**g**) RC-0C; (**h**) RC-10C; (**i**) RC-20C.

**Figure 12 materials-15-03099-f012:**
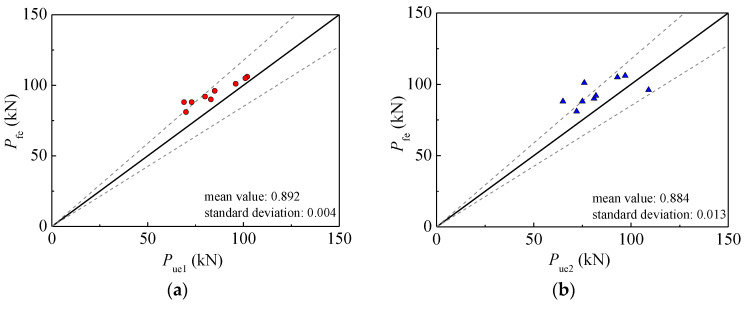
Comparison between *P*_fe_ and *P*_ue1_, *P*_ue2_ values of the specimens in Refs. [57,58,59]. (**a**) *P*_fe_/*P*_ue1_; (**b**) *P*_fe_/*P*_ue2_.

**Figure 13 materials-15-03099-f013:**
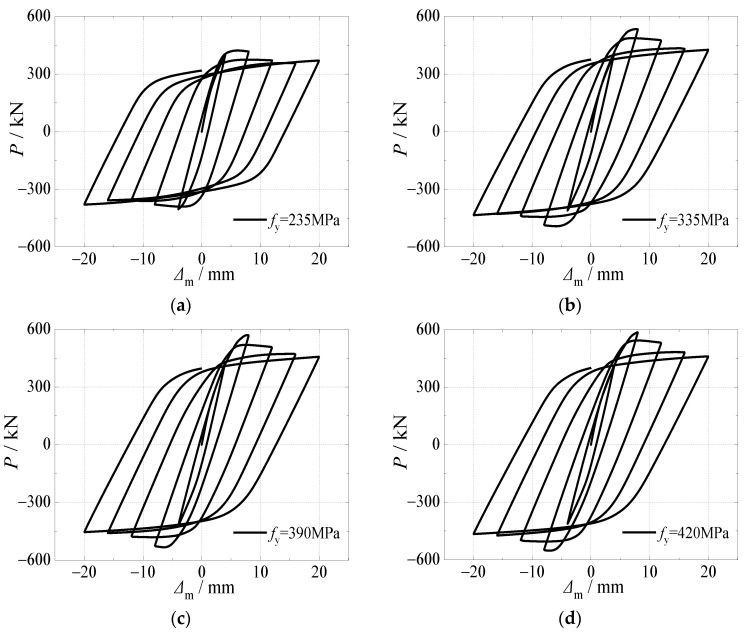
Effect of yield strength of steel tube on hysteretic curve. (**a**) *f*_y_ = 235 MPa; (**b**) *f*_y_ = 335 MPa; (**c**) *f*_y_ = 390 MPa; (**d**) *f*_y_ = 420 MPa.

**Figure 14 materials-15-03099-f014:**
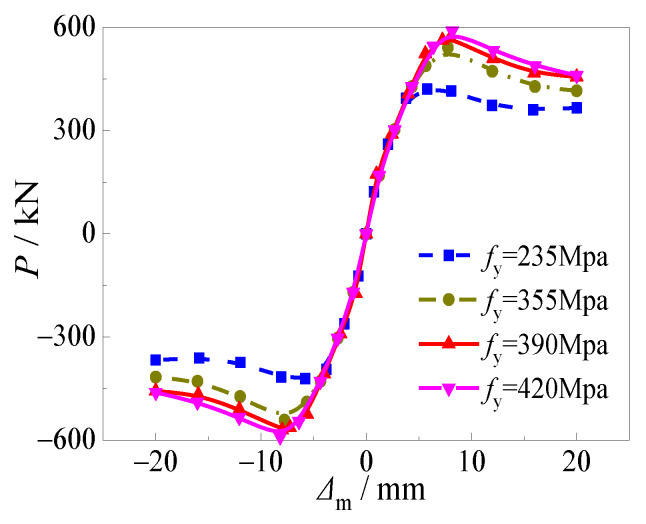
Effect of yield strength of steel tube on skeleton curve.

**Figure 15 materials-15-03099-f015:**
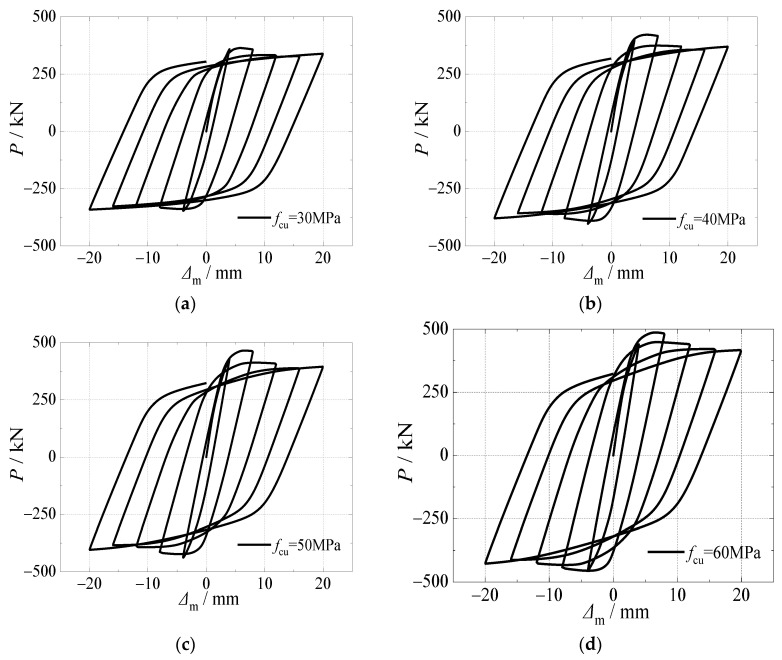
Effect of compressive strength of core concrete on hysteretic curve. (**a**) *f*_cu_ = 30 MPa; (**b**) *f*_cu_ = 40 MPa; (**c**) *f*_cu_ = 50 MPa; (**d**) *f*_cu_ = 60 MPa.

**Figure 16 materials-15-03099-f016:**
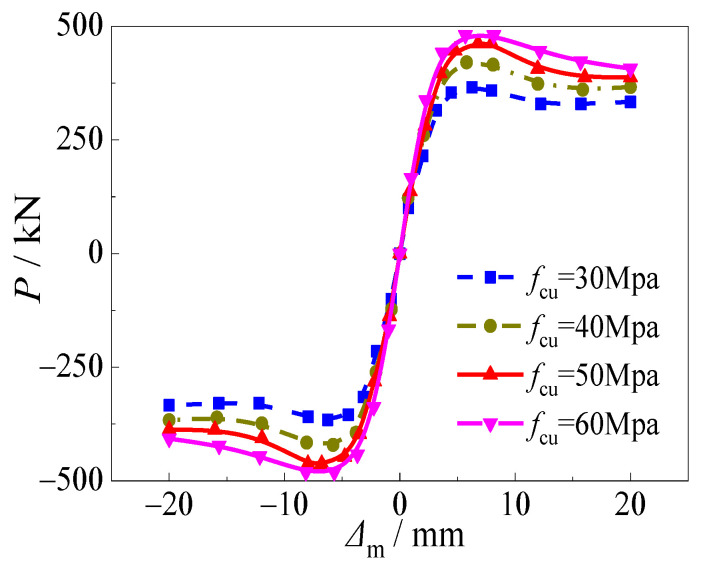
Effect of compressive strength of core concrete on skeleton curve.

**Figure 17 materials-15-03099-f017:**
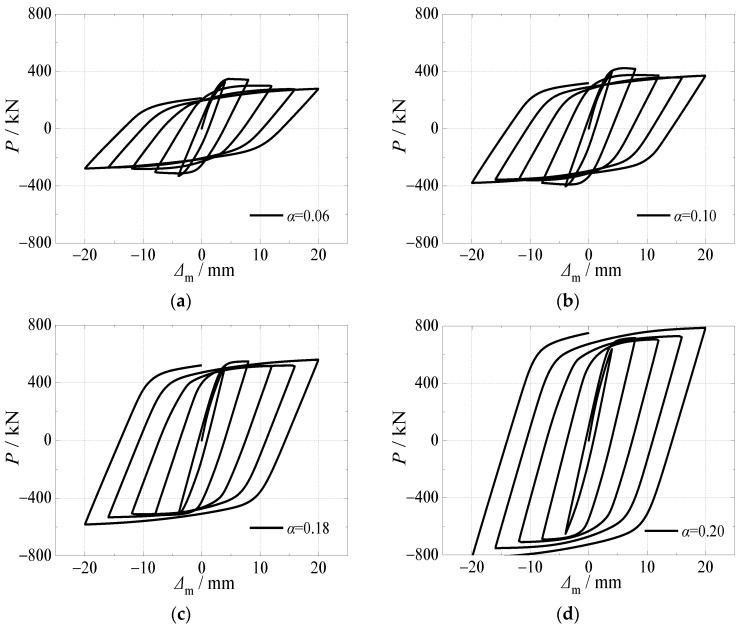
Effect of steel ratio on hysteretic curve. (**a**) *α* = 0.06; (**b**) *α* = 0.10; (**c**) *α* = 0.18; (**d**) *α* = 0.20.

**Figure 18 materials-15-03099-f018:**
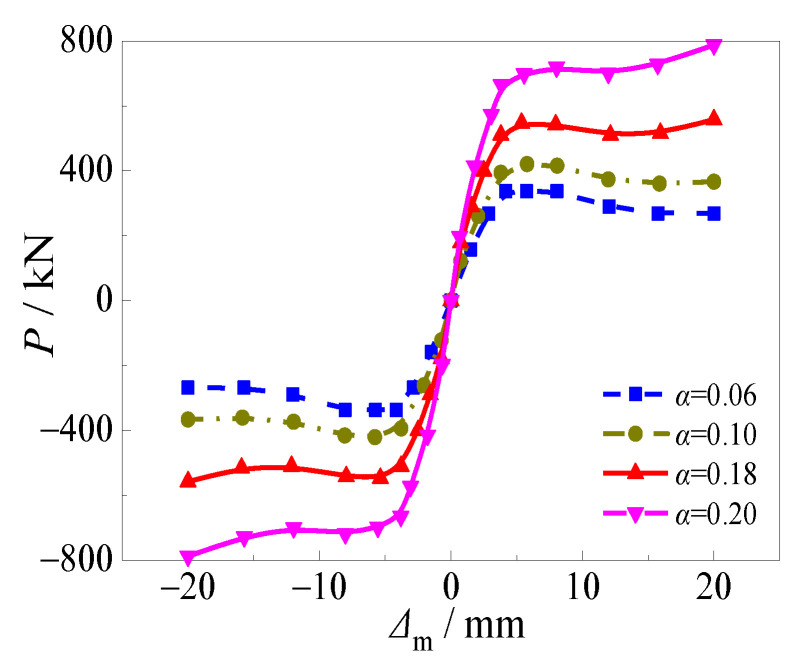
Effect of steel ratio on skeleton curve.

**Figure 19 materials-15-03099-f019:**
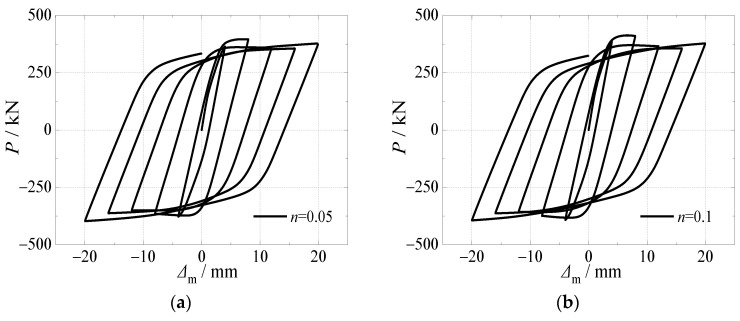
Effect of axial load ratio on hysteretic curve. (**a**) *n* = 0.05; (**b**) *n* = 0.10; (**c**) *n* = 0.15; (**d**) *n* = 0.20; (**e**) *n* = 0.25; (**f**) *n* = 0.30.

**Figure 20 materials-15-03099-f020:**
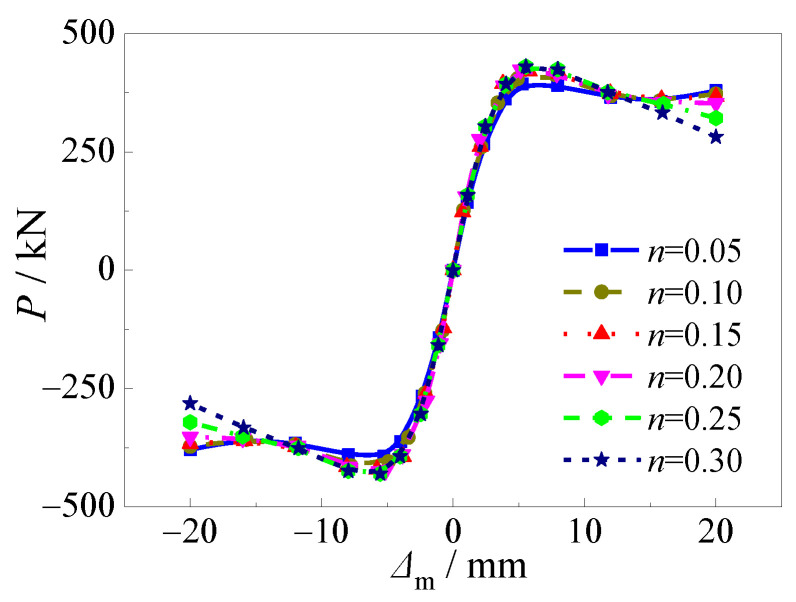
Effect of axial load ratio on skeleton curve.

**Figure 21 materials-15-03099-f021:**
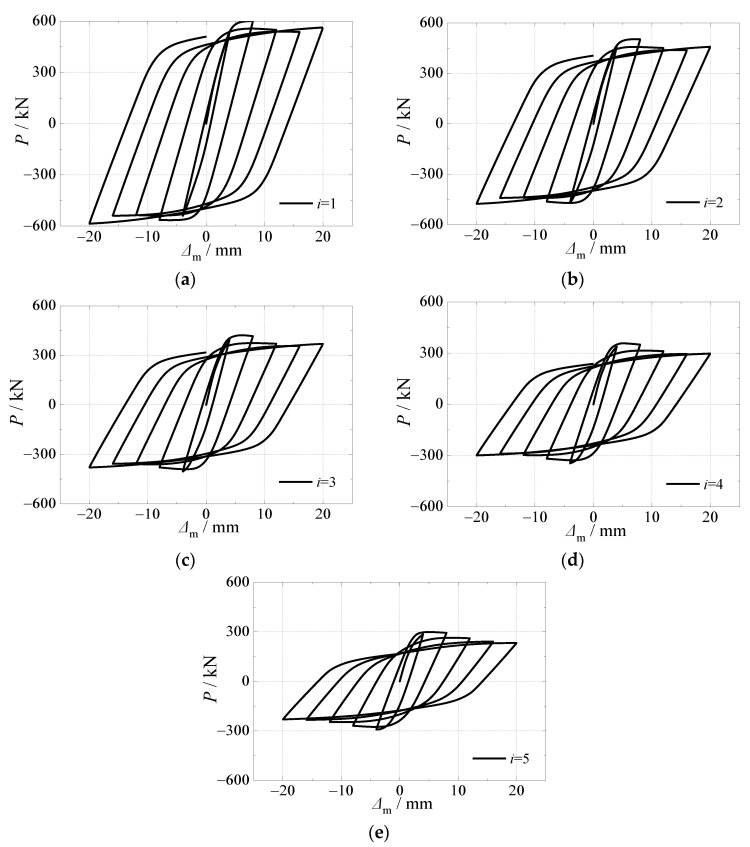
Effect of alternation time on hysteretic curve. (**a**) *i* = 1; (**b**) *i* = 2; (**c**) *i* = 3; (**d**) *i* = 4; (**e**) *i* = 5.

**Figure 22 materials-15-03099-f022:**
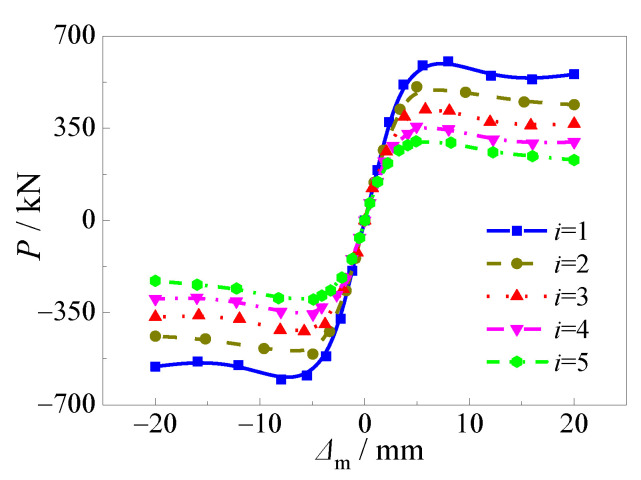
Effect of alternate times on skeleton curve.

**Figure 23 materials-15-03099-f023:**
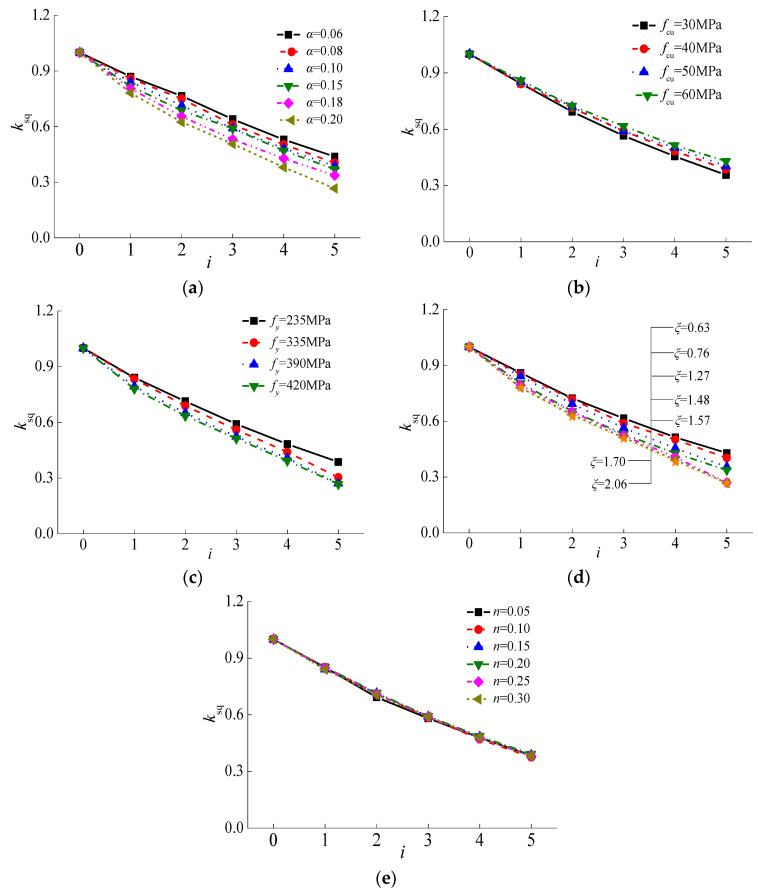
Effects of main parameters on reduction factor of horizontal bearing capacity. (**a**) Steel ratio; (**b**) Compressive strength of concrete; (**c**) Yield strength of steel tube; (**d**) Confinement coefficient; (**e**) Axial compression ratio.

**Figure 24 materials-15-03099-f024:**
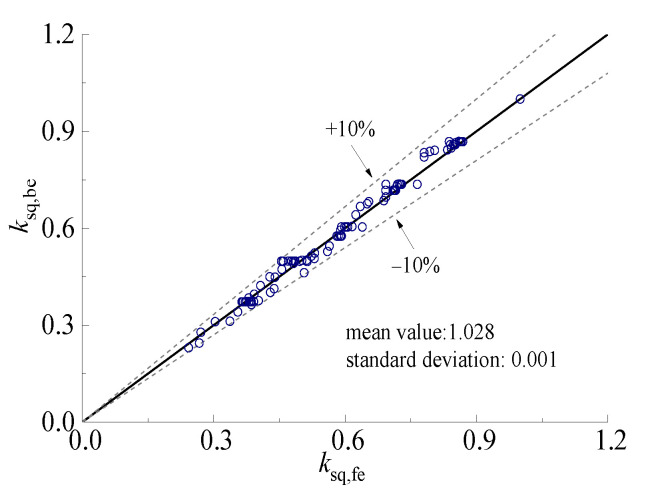
Comparison between calculated and simulated values of horizontal reduction coefficient.

**Figure 25 materials-15-03099-f025:**
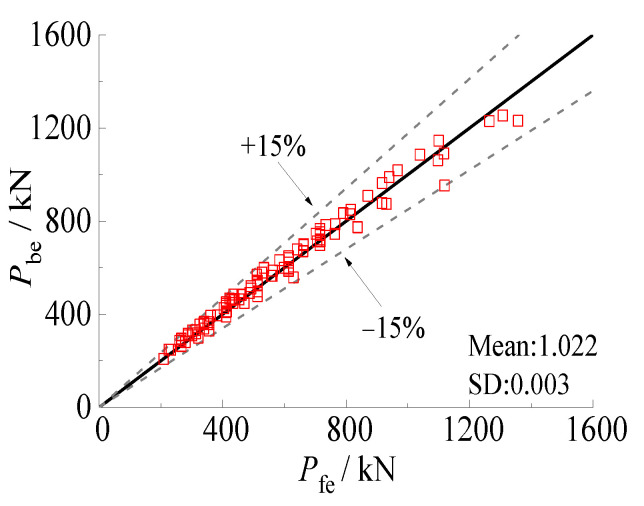
Comparison of ultimate horizontal load between calculated values and numerical values.

**Table 1 materials-15-03099-t001:** Specifications of the tested CFST columns in Refs. [34,35,36,37,38,39,40,41,42,43].

No.	Reference	Labels	*B* × *L* × *t* (mm)	*f*_cu, test_ (MPa)	*f*_y_ (MPa)	*f*_by_ (MPa)	*d* (mm)	*n*
1	[34,35]	STRC-60-1.5-4	200 × 600 × 1.89	60.0	309.2	357.0	16.4	0.40
2		STRC-60-1.5-6						0.60
3	[36,37]	STRC-70-4-1.5	200 × 600 × 3.00	75.1	254.0	449.4	11.0	0.35
4		STRC-70-5-1.5						0.45
5		STRC-70-6-1.5						0.55
6	[38]	FGZ4	250 × 1250 × 5.00	33.0	389.0	NA	NA	0.24
7	[39,40,41]	A1	200 × 600 × 6.00	50.1	330.0			0.20
8		A2						0.40
9		A3						0.60
10	[42,43]	RS-4L	250 × 530 × 4.00	41.0	265.0			0.20
11		RS-6M	250 × 530 × 6.00		317.6			0.40
12		RS-10H	250 × 530 × 10.00		242.2			0.54

Note: NA denotes the abbreviation of not available.

**Table 2 materials-15-03099-t002:** Comparison of ultimate horizontal load between specimens and FE model.

No.	Ref.	Specimen Labels	*P*_ue1_ (kN)	*P*_ue2_ (kN)	*P*_fe1_ (kN)	*P*_fe2_ (kN)	*P*_ue1_/*P*_fe1_	*P*_ue2_/*P*_fe2_
1	[34,35]	STRC-60-1.5-4	304	296	335	335	0.907	0.884
2		STRC-60-1.5-6	337	348	307	307	1.098	1.134
3		STRC-70-4-1.5	262	267	260	260	1.008	1.027
4	[36,37]	STRC-70-5-1.5	285	290	258	258	1.105	1.124
5		STRC-70-6-1.5	324	329	426	426	0.761	0.772
6	[38]	FGZ4	216	240	195	195	1.108	1.231
7		A1	397	275	433	433	0.917	0.635
8	[39,40,41]	A2	454	424	412	412	1.102	1.029
9		A3	445	449	436	436	1.021	1.030
10		RS-4L	434	361	458	458	0.948	0.788
11	[42,43]	RS-6M	556	572	544	544	1.022	1.051
12		RS-10H	773	771	614	614	1.259	1.256
Mean							1.064	1.022
SD							0.046	0.032

Note: Specimen labels in the table are the same as the test specimen in Refs. [34,35,36,37,38,39,40,41,42,43].

**Table 3 materials-15-03099-t003:** Main parameters of CFST pier columns in FE modeling.

Type	Parameter Value	Default Value
*α*	0.06	0.10
	0.10	
	0.18	
	0.20	
*f*_cu_ (MPa)	30	40
	40	
	50	
	60	
*f*_y_ (MPa)	235	235
	335	
	390	
	420	
*n*	0.05	0.15
	0.10	
	0.15	
	0.20	
	0.25	
	0.30	
*i*	0	3
	1	
	2	
	3	
	4	
	5	

## Data Availability

The study did not report any data.

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
