# Peer review of "Analytical Hysteretic Behavior of Square Concrete-Filled Steel Tube Pier Columns under Alternate Sulfate Corrosion and Freeze-Thaw Cycles"

_materials, 2022, doi:10.3390/ma15093099_

Round 1

Reviewer 1 Report

The paper entitled “Analytical hysteretic behavior of square concrete-filled steel tube pier columns under alternate of sulfate corrosion and freeze-thaw cycles” presents the results of a parametric study carried out by using a FE model calibrated on the basis of experimental tests. Interesting results have been obtained in terms of hysteresis loops. In addition, some design formula are proposed.

  The manuscript presents a topic of scientific interest but there are issues that need to be addressed before considering it for publication:   1) In the abstract, the expression “pinch phenomenon” should be substituted by “pinching phenomenon”.    2) The following keyword “concrete-filled steel tube pier column” is too long. Try to shorten it.   3) In the Introduction Section (page 2, line 48), the expression “energy consumption” is not formally correct. Do you mean “energy dissipation”? Please, substitute such an expression that has been adopted also in other parts of the manuscript.   4) In the Introduction Section (page 2, line 61), the expression “earthquake waves” should be substituted by “earthquake excitations”.   5) In Section 2 (page 3, line 123), the following sentence is not clear: “The test results were reported in ref were adopted….”. Please, try to make it clearer. In addition, at line 135, the expression “could be experimented” is not correct from a grammar point of view.   6) In Section 2 (page 4), the sentence at line 150 to 152 is not clear at all. Please, revise it.   7) In Section 3 (page 5), the following sentence “Modeling hysteretic behavior of square CFST is determined by three main parts” needs to be substituted by the more precise following one: “The phenomenological modeling of square CFSTs, typically performed by using differential [A,B] or non-differential models [C,D], is determined by three main parts”. [A] https://doi.org/10.1061/JMCEA3.0002106 [B] https://doi.org/10.1115/1.3153594 [C] https://doi.org/10.1007/s11071-018-4282-2 [D] https://doi.org/10.1016/j.ymssp.2020.106984 8) In Section 3 (page 5, line 190), the expression “hysteretic load” does not have any sense. We may have a hysteretic behavior but not a hysteretic load. 9) The authors have adopted the simplest hysteretic model that can be used, that is, the bilinear model. It is necessary to remember in the manuscript that much more sophisticated and accurate models are actually available in the literature [E, F]. [E] https://doi.org/10.1016/j.jsv.2008.01.018 [F] https://doi.org/10.1007/s11071-019-05022-5   10) It is not possible to understand the correlation between line 211 and Equation (3). What do all the parameters at line 211 refer to? 11) In Section 4, it is necessary to correct: a) Fig. 15 and Fig. 16 depicts, b) Fig. 17 and Fig. 18 depicts, c) and so on. 12) In the conclusion section, the expression “in simulate” needs to be substituted by “in simulating”.

Author Response

Response to Reviewers’ comments

Manuscript Number: materials-1674627

Title: Analytical hysteretic behavior of square concrete-filled steel tube pier columns under alternate sulfate corrosion and freeze-thaw cycles

Authors: Tong Zhang, Qianxin Wen, Lei Gao, Qian Xu*, Jupeng Tang

The authors would like to thank the Reviewers for their comments. The revised version of manuscript addresses the comments of reviewers as detailed in the following. Parts modified have been marked in yellow in the resubmission for ease of reference.

-Reviewer 1

The paper entitled “Analytical hysteretic behaviour of square concrete-filled steel tube pier columns under alternate of sulfate corrosion and freeze-thaw cycles” presents the results of a parametric study carried out by using a FE model calibrated on the basis of experimental tests. Interesting results have been obtained in terms of hysteresis loops. In addition, some design formula are proposed.

The manuscript presents a topic of scientific interest but there are issues that need to be addressed before considering it for publication:

Comment 1: In the abstract, the expression “pinch phenomenon” should be substituted by “pinching phenomenon”

Response: Thank you for this comment. We corrected the expression of this sentence in line 25 in the revised manuscript as requested.

“The compressive strength of concrete and alternation time significantly decreased the unloading stiffness which causes the pinching phenomenon.”

Comment 2: The following keyword “concrete-filled steel tube pier column” is too long. Try to shorten it.

Response: Thank you for this comment. We shorten the keyword “concrete-filled steel tube pier column” as “CFST pier column”.

Comment 3: In the Introduction Section (page 2, line 48), the expression “energy consumption” is not formally correct. Do you mean “energy dissipation”?. Please, substitute such an expression that has been adopted also in other parts of the manuscript.

Response: Thank you for this comment. We corrected the expression of this phrase in lines 48-51 and lines 254-257 in the revised manuscript as requested.

“Tomii M [14] firstly studied the seismic behaviour of square CFST through quasi-static method and illustrated that the CFST was excellent in energy dissipation.”

“Based on the characteristics of energy absorption and energy dissipation in the process of plastic deformation…”

Comment 4: In the Introduction Section (page 2, line 61), the expression “earthquake waves” should be substituted by “earthquake excitations”.

Response: Thank you for this comment. We corrected the expression of this sentence in lines 63-65 in the first revised manuscript as requested.

“It illustrated that the vertex displacement of pier columns was below the ultimate displacement for all 18 earthquake excitations…”

Comment 5: In Section 2 (page 3, line 123), the following sentence is not clear. “The test results were reported in ref were adopted…”. Please, try to make it clear. In addition, at line 135, the expression “could be experimented” is not correct from a grammar point of view.

Response: Thank you for this comment. We rewrote this sentence in lines 132-134 and 146-147 in the first revised manuscript as requested to make it clear.

(1) “The test results reported in previous studies [34-43] were selected to verify the CFST pier columns modelled in ABAQUS software [44].”

(2) “Fig. 1. presented details of the typical testing device, the hysteretic behavior of the specimens could be tested.”

Comment 6: In Section 2 (page 4), the sentence at line 150 to 152 is not clear at all. Please, revise it.

Response: Thank you for this comment. We rewrote the sentences in lines 157-164 in the first revised manuscript as requested.

“In the force control phase, one cycle was imposed at the load levels of 0.5Nvy, 0.5Nvy, and Nvy, respectively, where Nvy is the estimated value of specimen yield strength. Afterward, the displacement-controlled loading was performed at the incremental levels of 1δy, 2δy, 3δy, etc, where δy presents the estimated displacement corresponding to the yield strength, and the cyclic loading was repeated twice for each displacement level. The yield strength Nvy and yield displacement δy were tested by GYM (general yield moment) method.”

Comment 7: In section 3 (page 5), the following sentence “Modeling hysteretic behavior of square CFST is determined by three main parts” needs to be substituted by the more precise following one: “The phenomenological modeling of square CFSTs, typically performed by using differential [A,B] or non-differential models [C,D], is determined by three main parts.”

[A] https://doi.org/10.1061/JMCEA3.0002106

[B] https://doi.org/10.1115/1.3153594

[C] https://doi.org/10.1007/s11071-018-4282-2

[D] https://doi.org/10.1016/j.ymssp.2020.106984

Response: Thank you for this comment. We rewrote the expression of this sentence in lines 175-176 in the revised manuscript as requested.

“The phenomenological modeling of square CFSTs typically performed by using differential [52,53] or non-differential models [54,55], is determined by three main parts.”

[52] Wen YK. Equivalent linearization for hysteretic systems under random excitation. Journal of Applied Mechanics, 1980; 47:150-154.

https://doi.org/10.1061/JMCEA3.0002106

[53] Wen Y. K. Equivalent linearization for hysteretic systems under random excitation. Journal of Applied Mechanics, 1980, 47:150-154.

https://doi.org/10.1115/1.3153594

[54] Vaiana Nicolo, Sessa Salvatore, Marmo Francesco, Rosati Luciano. A class of uniaxial phenomenological models for simulating hysteretic phenomena in rate-independent mechanical systems and materials. Nonlinear dynamics, 2018, 93(3):1647-1669.

https://doi.org/10.1007/s11071-018-4282-2

[55] Vaiana Nicolo, Sessa Salvatore, Rosati Luciano. A generalized class of uniaxial rate-independent models for simulating asymmetric mechanical hysteresis phenomena. Mechanical Systems and Signal processing, 2021, 146: 106984.

https://doi.org/10.1016/j.ymssp.2020.106984

Comment 8: In Section 3 (page 5, line 190), the expression “hysteretic load” does not have any sense. We may have a hysteretic behaviour but not a hysteretic load.

Response: Thank you for this comment. We corrected the expression of this sentence in lines 209-211 in the first revised manuscript as requested.

“The stress-strain skeleton curve of steel tube corresponding to the hysteretic behaviour is composed of two parts, which are elastic stage and strain hardening stage as showed in Fig 4.”

Comment 9: The authors have adopted the simplest hysteretic model that can be used, that is, the bilinear model. It is necessary to remember in the manuscript that much more sophisticated and accurate models are actually available in the literature [E, F].

[E] https://doi.org/10.1016/j.jsv.2008.01.018

[F] https://doi.org/10.1007/s11071-019-05022-5

Response: Thank you for this comment. We rewrote the sentence in section 3.1.1 in lines 203-208 in the first revised manuscript and added description of other advanced and sophisticated hysteretic model as requested. The authors were inspired by these references suggested by the reviewer and would use these models in our later investigation.

“Bouc-Wen modified model [56] is not particularly sensitive to the selection of its parameter. Vaiana et al [57] proposed a hysteretic mechanical system by combining a novel rate-independent model and an explicit time integration method which avoid the convergence problem for its small time step. Since the hysteretic behaviors of steel tube under cyclic loading are complex and the effect of corrosion-freeze-thaw cycles has strong randomness, using bilinear model is more easy to describe the complex environment effect.”

[56] Charalampakis A E, Koumousis V K. Identification of Bouc-Wen hysteretic systems by a hybrid evolutionary algorithm. Journal of Sound and Vibration, 2008, 314: 571-585.

[57] Vaiana Nicolo, Sessa Salvotore, Marmo Francesco, Rosati Luciano. Nonlinear dynamic analysis of hysteretic mechanical systems by combining a novel rate-independent model and an explicit time integration method. Nonlinear dynamics, 2019: 05022-5.

Comment 10: It is not possible to understand the correlation between line 211 and Equation (3). What do all the parameters at line 211 refer to?

Response: Thank you for this comment. We defined the parameters in Equation (3) with more details in lines 231-239 in the first revised manuscript as requested.

“where, x=ε/ε0, ε is the strain of the concrete and ε0 is the peak strain corresponding to peak load, ε0=(1300+12.5fc+800ξe2)×10-6; y=σ/σ0, σ0=fc(1-0.035Nft/100), σ is the stress of concrete and σ0 is the stress corresponding to the peak load, fc=[0.76+0.2log10(fcu/19.6)]fcu, fc is the cylinder strength of concrete, fcu is the cube compressive strength of concrete, Nft is the axial load of specimens after corrosion-freeze-thaw cycles, η=1.6+1.5/x, η is defined as shape coefficient to reflect the properties of shape of cross section, β=(fc)0.1/[1.2(1+ξe)0.5], ξe =αe×fye/fck, ξe is the effective confinement factor, fye and fck are the effective yield strength of steel tube and characteristic compressive strength of concrete, respectively, fck=0.88×0.76×fcu, αe is steel ratio, αe=Ase/Ac, Ase and Ac are the effective cross section area of the steel tube and core concrete, respectively.”

Comment 11: In Section 4, it is necessary to correct: a) Fig. 15 and Fig. 16 depicts, b) Fig. 17 and Fig. 18 depicts, c) and so on.

Response: Thank you for this comment. We corrected the expression of these sentence in lines 364-366, 385-386, 400-402, and 414-415 in the first revised manuscript as requested.

(1) “Fig. 15, and Fig. 16 depicted the effects of compressive strength of concrete…”

(2) “Fig. 17, and Fig. 18 depicted the variations of hysteretic curves and skeleton curves…”

(3) “Fig. 19, and Fig. 20 presented the effects of axial compression ratio on hysteretic curves and skeleton curves…”

(4) “Fig. 21 depicted the effects of alternate time on hysteretic curves of CFST piers under horizontal load…”

Comment 12: In the conclusion section, the expression “in simulate” needs to be substituted by “in simulating”.

Response: Thank you for this comment. We corrected the expression of this sentence in lines 526-529 in the first revised manuscript as requested.

“The nonlinear finite element (FE) model considering the dual environmental impact of sulfate corrosion and freeze-thaw cycles was accurate in simulating the degradation of stiffness of the loading stiffness and unloading stiffness, bearing load and ductility on the basis of reasonable damage factors.”

Reviewer 2 Report

This paper has an interesting subject on the combined effect of sulfate corrosion and freeze-thaw cycles on square concrete-filled steel tube pier columns.  The following changes and improvements of this paper should be performed.

  • The whole text should be carefully checked for grammar, syntax and the consistency/meaning of sentences (including the title of this paper).  In the Introduction, please explain, where the expressions "it demonstrated", "it found" etc. refer to.   Please check the subscripts of the chemical types mentioned in text (e.g. SO"42" ?, etc)
  • In the abstract, please explain better "the alternation time" and if this is connected with a loading type.   
  • In line 58, please explain "LRFD".  Similarly, please explain "7-degree" in line 60 (please make this clear for international readers)
  • Please add reference for used software, ABAQUS.
  • Please, explain about the mentioned specimens in lines 123-135, if these specimens comply with a specific code and give more relative details.  
  • Please, check subscripts of symbols and mathematical expressions in the whole text (e.g. lines 180-181, lines 211-212 etc). 
  • In equation 1 (lines 177-179), the letters are too small.  Also, this equation is based on literature reference or calculated by the authors?
  • In fig. 6, an axial load is mentioned.  Please refer to/explain the values of this axial load.  
  • Please explain the value range of "ambient temperature" (section 3.5.1 etc).
  • Please add some details on the considered sulfate corrosion type, to make the ordinar reader more convenient.
  • The mentioned codes in the experiments of this paper are mostly written in chinese, which cannot be read by all people.  Please, add  comment(s) about the used assumptions/codes available for international readers.  
  • In the conclusions, please explain more, even by range of values, the mentioned "positive/negative effect". 

Author Response

Response to Reviewers’ comments

Manuscript Number: materials-1674627

Title: Analytical hysteretic behavior of square concrete-filled steel tube pier columns under alternate sulfate corrosion and freeze-thaw cycles

Authors: Tong Zhang, Qianxin Wen, Lei Gao, Qian Xu*, Jupeng Tang

The authors would like to thank the Reviewers for their comments. The revised version of manuscript addresses the comments of reviewers as detailed in the following. Parts modified have been marked in yellow in the resubmission for ease of reference.

-Reviewer 2

This paper has an interesting subject on the combined effect of sulfate corrosion and freeze-thaw cycles on square concrete-filled steel tube pier columns. The following changes and improvements of this paper should be performed.

Comment 1: The whole text should be carefully checked for grammar, syntax and the consistency/meaning of sentences (including the title of this paper).

In the Introduction, please explain, where the expressions “it demonstrated”, “it found” etc. refer to. Please check the subscripts of the mechanical types mentioned in test (e.g. SO“42”?.etc)

Response: Thank you for this comment. We checked the manuscript and corrected the title of this paper. We rewrote the sentences to make the expression of “it demonstrated” and “it found” etc. clearer in lines 92-94, 96-98, 99-101, and 105-107. Then, we corrected the expression of sentences including SO42- in lines 84-89 in the first revised manuscript as requested.

(1) “Analytical hysteretic behavior of square concrete-filled steel tube pier columns under alternate sulfate corrosion and freeze-thaw cycles”

(2) “The test results demonstrated that the freeze-thaw cycle significantly influenced the load bearing capacity….”

“And Wang [27] found that the cyclic time has a negative linear relationship on load bearing capacity and initial stiffness of the specimens.”

“The test results presented that square specimens destroyed in crushing failure…”

“Gao et al. [31] found that the axial compressive strength decreased linearly with the cyclic time while increased with the higher compressive strength of concrete.”

(3) “Zhang [25] carried out a durability test on 20 thin-walled steel tube columns under SO42- corrosion. The study has found that ultimate bearing capacity, ductility, and stiffness decreased with the corrosion ratio, especially for the specimens with thinner wall. The finial failure mode of the CFST stub columns under sulfate corrosion belonged to shear failure. After the corrosion of SO42-, local buckling of the slender steel tube would occur to a large extent.”

Comment 2: In the abstract, please explain better “the alternation time” and if this is connected with a loading type.

Response: Thank you for this comment. We rewrote the sentence in lines 25-27 as follows:

“The yield strength of steel, compressive strength of concrete and alternation time of environmental factors (corrosion-freeze-thaw cycles) has no obvious effect on the initial stiffness while the steel ratio has a remarkable effect.”

Comment 3: In line 58, please explain “LRFD”. Similarly, please explain “7-degree” in line 60 (please make this clear for international readers)

Response: Thank you for this comment. We corrected the expression of these sentence in lines 58-60, and 60-63 in the first revised manuscript as requested.

(1) “Then, based on the simple plastic model, a new equation was proposed and the specification “Guide for Load and Resistance Factor Design (LRFD) Criteria for Offshore Structures” for the seismic design of bridge piers were recommended.”

(2) “Li [18] carried out a dynamic time-history analysis of CFST piers according to a practical engineering at 7-degree (M=0.58I+1.5, where M is Richter scale, I denotes seismic fortification intensity, I=7 here) protected earthquake intensity.”

Comment 4: Please add reference for used software, ABAQUS.

Response: Thank you for this comment. We added reference for the ABAQUS software in line 646 in the first revised manuscript as requested.

[44] ABAQUS, ABAQUS Standard User’s Manual. Version 6.13, Dassault Systemems Corp., Providence (RI, USA), 2013.

Comment 5: Please, explain about the mentioned specimens in lines 123-135, if these specimens comply with a specific code and give more relative details.

Response: Thank you for this comment. We rewrote the expression of these sentences and added details in lines 136-143 in the revised manuscript as requested.

“As showed in Table 1, the length of the specimens ranged from 530 to 1250 mm corresponding to the length-to-width (L/B) ratio of 2.12-5.00. Thus, influence of slenderness effect could be avoided [45]. The compressive strength of concrete core ranges from 33.0 to 75.1 MPa, which contains normal strength concrete and high strength concrete [46] (fck≤50MPa is defined as normal strength concrete). Specimens with reinforcement bars consist 8Φ16 and 4Φ11, and the ties are used Φ8@100 mm and Φ8@200 mm, which meet the specification requirements [46].

Comment 6: Please, check subscripts of symbols and mathematical expressions in the whole text (e.g. lines 180-181, lines 211-212 etc.).

Response: Thank you for this comment. We corrected the symbols and mathematical expression in the test in lines 193-194, 231-239, 242-243, and 248-249 in the first revised manuscript as requested.

(1) “where εe = 0.8fye/Ese, εe1 = 1.5εe, εe2 = 10εe1, εe3 = 100εe1, A = 0.2fy(εe1-εe)2, B = 2e1, C = 0.8fye+e2-e, Ese = (1-0.525γ)Es, fye = (1-0.908γ)fy, and β = Δt/t, Δt = t-te.”

(2) “where σ is stress of steel tube, Ese is the elastic modulus of steel, ε is the strain of steel tube below yield strain, εy is the yield strain corresponding to yield strength of steel tube.”

(3) “where, x=ε/ε0, ε is the strain of the concrete and ε0 is the peak strain corresponding to peak load, ε0=(1300+12.5fc+800ξe2)×10-6; y=σ/σ0, σ0=fc(1-0.035Nft/100), σ is the stress of concrete and σ0 is the stress corresponding to the peak load, fc=[0.76+0.2log10(fcu/19.6)]fcu, fc is the cylinder strength of concrete, fcu is the cube compressive strength of concrete, Nft is the axial load of specimens after corrosion-freeze-thaw cycles, η=1.6+1.5/x, η is defined as shape coefficient to reflect the properties of shape of cross section, β=(fc)0.1/[1.2(1+ξe)0.5], ξe =αe×fye/fck, ξe is the effective confinement factor, fye and fck are the effective yield strength of steel tube and characteristic compressive strength of concrete, respectively, fck=0.88×0.76×fcu, αe is steel ratio, αe=Ase/Ac, Ase and Ac are the effective cross section area of the steel tube and core concrete, respectively.”

(4) “where Ec is elastic modulus of core concrete, fc denotes the cylinder strength of concrete.”

(5) “where ξe is the effective confinement factor, fcis the cylinder strength of concrete, fb0 is the initial biaxial strength of concrete, fc0 is the initial axial strength of concrete.”

Comment 7: In equation 1 (lines 177-179), the letters are too small. Also, this equation is based on literature reference or calculated by the authors?

Response: Thank you for this comment. We corrected the size of Eq. (1) in lines 184-186 in the first revised manuscript as requested. The equation was based on the five-stage model, and modified according to the effect sulfate corrosion and freeze-thaw cycles on the relationship of stress-strain.

“A secondary plastic flow stress-strain curve [45] was modified according to the effect of corrosion-freeze-thaw cycles on the relationship of stress-strain and then adopted to define the material behaviour…”

(1)

Comment 8: In fig. 6, an axial load is mentioned. Please refer to/explain the values of this axial load.

Response: Thank you for this comment. We corrected the expression of axial load in Fig. 6 as “N” in lines 282-283 in the first revised manuscript as requested. In this paper, one main parameter axial compression ratio n is related to axial load. (n=N/N0, where N is axial load of column under axially load, N0 is ultimate compressive strength of CFST calculated by equation N0=fcAc+fyAs, As and Ac are the area of steel tube and core concrete respectively; fc and fs is the nominal prism strength of concrete and yield strength of steel tube.)

Figure 6. Schematic views of FE meshing and boundary conditions.

Comment 9: Please explain the value range of “ambient temperature” (section 3.5.1 etc.).

Response: Thank you for this comment. We explained the value range of “ambient temperature” in lines 313-314 in the first revised manuscript as requested.

“Typical failure mode of specimens under cyclic loading at ambient temperature (20℃±2℃) was depicted in Fig. 9.”

Comment 10: Please add some details on the considered sulfate corrosion type, to make the ordinar reader more convenient.

Response: Thank you for this comment. We added the details on the considered sulfate corrosion type in lines 127-134 in the first revised manuscript as requested.

“The square CFST pier columns was corroded through electrochemical corrosion method. The electrolyte was mixed solution of Ca(NO3)2, Na2SO4, and NH4Cl, and adopted HNO3 to adjust pH value which was set as 4.5. The specimens were fully immersed into the electrolyte. The corrosion rate was defined as the loss of mass of the steel and corrosion rate γ=10%, 20%, and 30% corresponding to the terms of service of 5, 10, and 15 years. The hemispherical pitting was formed on the surface of steel tube.”

Comment 11: The mentioned codes in the experiments of this paper are mostly written in Chinese, which cannot be read by all people. Please, add comment(s) about the used assumptions/codes available for international readers.

Response: Thank you for this comment. We rewrote the expression of this sentence in lines 144-146, and 156-164 in the first revised manuscript as requested. And the authors added some details of hysteretic experiment.

(1) “The materials properties of steel and infilled concrete could be obtained from test or specifications such as GB50010 [46], ACI-318-08 [47], EN-1992-1-1 [48], and GB20017 [49], AS4100 [50], etc.”

(2) “According to JGJ/T101-2015 [51], the loading history of cyclic loading included a force control phase and a displacement control phase, as showed in Fig.2. In the force control phase, one cycle was imposed at the load levels of 0.5Nvy, 0.5Nvy, and Nvy, respectively, where Nvy is the estimated value of specimen yield strength. Afterward, the displacement-controlled loading was performed at the incremental levels of 1δy, 2δy, 3δy, etc, where δy presents the estimated displacement corresponding to the yield strength, and the cyclic loading was repeated twice for each displacement level. The yield strength Nvy and yield displacement δy were tested by GYM (general yield moment) method.”

Comment 12: In the conclusions, please explain more, even by range of values, the mentioned “positive/negative effect”.

Response: Thank you for this comment. We rewrote the conclusion mentioned “positive/negative effect” in lines 526-529, 538-542, 544-548, and 551-554 in the first revised manuscript as requested.

(1) “The cyclic curves of hysteresis loops increased with increasing steel ratio and yield strength of steel tube caused the increasing initial stiffness of the specimens. The skeleton curves corresponding to the hysteresis loops presented ductile beahvior with increasing steel ratio and yield strength of steel tube…The ultimate horizontal load increased by 21.4%-126.4%, 26.3%-39.2%, and 16.0%-33.7% with the steel ratio increasing from 0.06 to 0.20, yield strength of steel tube increasing from 235 MPa to 420 MPa, compressive strength of core concrete increasing from 30 MPa to 60 MPa, respectively while the ultimate horizontal load decreased by 16.2%-50.2% with alternation time increased from 1 time to 5 times.”

(2) “The ductility of the specimens increased with the increasing steel ratio since the post stage of skeleton curve varied from descending stage to ascending stage, while the ductility of specimens decreased with the increasing compressive strength, axial compression ratio, and alternation time for the increasing brittleness caused by the damage of raw materials.”

(3) “The horizontal bearing capacity decreased by 16.2%-50.2% with the alternation time increasing from 1time to 5 times. And the horizontal bearing capacity decreased up to 17.7% with the confinement coefficient increased from 0.63 to 2.06.”

Round 2

Reviewer 1 Report

The authors made the required revisions. The manuscript can be accepted.

Reviewer 2 Report

The authors have answered the questions of the reviewers.  This article can be published.